



# A comprehensive verification of the weather radar-based hail metrics POH and MESHS and a recalibration of POH using dense crowdsourced observations from Switzerland

Jérôme Kopp[1], Alessandro Hering[2], Urs Germann[2], and Olivia Martius[1]

[1]Oeschger Centre for Climate Change Research and Institute of Geography, University of Bern, Bern, Switzerland
[2]Federal Office of Meteorology and Climatology MeteoSwiss, Locarno-Monti, Switzerland

**Correspondence:** Jérôme Kopp (jerome.kopp@unibe.ch)

**Abstract.** Remote hail detection and hail size estimation using weather radar observations has the advantage of wide spatial coverage and high spatial and temporal resolution. Switzerland National Weather Service (MeteoSwiss) uses two radar-based hail metrics: the probability of hail at the ground (POH) to assess the presence of hail, and the maximum expected severe hailstone size (MESHS) to estimate the largest hailstone diameter. However, radar-based metrics are not direct measurements of hail and have to be calibrated with and verified against ground-based observations of hail, such as crowdsourced hail
reports. Switzerland benefits from a particularly rich and dense dataset of crowdsourced hail reports from the MeteoSwiss app. We combine a new spatiotemporal clustering method (ST-DBSCAN) with radar reflectivity to filter the reports and use the filtered reports to verify POH and MESHS in terms of the Hit Rate, False Alarms Ratio (FAR), Critical Success Index (CSI), and Heidke Skill Score (HSS). Using a 4 km × 4 km maximum upscaling approach, we find FAR values between 0.3 and 0.7 for POH and FAR > 0.6 for MESHS. For POH, the highest CSI (0.37) and HSS (0.52) are obtained for a 60% threshold, while
for MESHS the highest CSI (0.25) and HSS (0.4) are obtained for a 2 cm threshold. We find that the current calibration of POH does not correspond to a probability and suggest a recalibration based on the filtered reports.

## 1 Introduction

Remote hail detection and hail size estimation using weather radar observations is done operationally in several countries (e.g., Nisi et al., 2016; Punge and Kunz, 2016; Allen et al., 2020). A big advantage of using radar observations is their wide spatial coverage and their high spatial and temporal resolution (Punge and Kunz, 2016; Allen et al., 2020). Two single-polarisation radar-based hail metrics are operationally used in Switzerland by the national weather services MeteoSwiss: i) the Probability Of Hail (POH; Foote et al., 2005) based on the Waldvogel criteria (Waldvogel et al., 1979) and its subsequent link to a
probability (Witt et al., 1998) is used to estimate the presence of hail of any size at the ground (operational since 2008); and ii) the Maximum Expected Severe Hailstone Size (MESHS; Joe et al., 2004) based on (Treloar, 1998) is used to estimate the





maximum hailstone size at the ground, for hailstone diameters equal to or larger than 2 cm (operational since 2009). Both metrics are based on the height difference between the highest altitude at which a certain radar reflectivity is measured (45 dBZ for POH, 50 dBZ for MESHS) and the 0°C altitude, which is a proxy for the area where hail can grow by collecting

supercooled water droplets (Doswell, 2001; Allen et al., 2020).

The radar-based hail metrics form the basis of the Swiss hail climatology (NCCS, 2021) and they were used to study hail variability and hail storm characteristics (e.g. Nisi et al., 2016, 2018; Madonna et al., 2018; Schmid et al., 2023) and they are used by insurance companies for damage assessments. Moreover, they were recently used as target variables in a deep learning model for thunderstorm prediction (Leinonen et al., 2023) and for the improvement of the operational MeteoSwiss

thunderstorms nowcasting and warning system TRT (Hering et al., 2004, 2022).

However, those radar-based metrics are proxies and not direct measurements of hail on the ground. Consequently, they have to be initially calibrated with and further verified against ground-based observations of hail. Ground-based observations are challenging to gather because of the nature of hail: for example in Switzerland, hailstorms occur 2 to 4 times per square kilometer per year in the hail hotspots (NCCS, 2021) and typically last less than ten minutes locally (Kopp et al., 2023). Con-

sequently, calibration and verification of radar-based hail algorithms are based on a limited number of observations. Waldvogel et al. (1979) used 195 storm cells in Switzerland, among which only 33 were hail-producing cells to verify their criteria. They found that the criteria detected all hail cells (a 100% hit rate) but that half of the identified cells never produced hail (a 50% false alarm ratio). Treloar (1998) used 27 hailstorms in the area of Sydney to propose the initial version of MESHS and Joe et al. (2004) verified it with a single day of data in Australia. While to our knowledge MESHS was only validated against

observations in Joe et al. (2004) and Barras et al. (2019), several versions of the Waldvogel criteria and of the POH metric have been validated in various countries (e.g. Kessinger et al., 1995; Holleman, 2001; Kunz and Kugel, 2015; Nisi et al., 2016; Puskeiler et al., 2016; Barras et al., 2019). Pooled together, those studies gave more robust results but still had two limitations.

First, all of them used a distance buffer to match the actual location of an observation and the radar detection. Kessinger et al. (1995) used a 15 km influence region consistent with the underlying storm definition process. Holleman (2001) chose a

12.5 km positioning tolerance, Kunz and Kugel (2015) a 10 km grid cell and Puskeiler et al. (2016) a 5 km × 5 km area around the radar grid point, both to account for the spatial resolution of their observations (at the district or postal code level). Car insurance reports used by Nisi et al. (2016) needed to focus on large urban areas of Switzerland. The use of a physically based distance buffer is necessary to account for the potential wind drift of hail (up to 3 km according to Hohl et al., 2002; Schuster et al., 2006). However, using significantly larger values can artificially increase the actual hit rate H and reduce the actual FAR.

The second limitation is related to the nature of the observations; often, insurance damage claims were used as ground truth for the validation (Holleman, 2001; Hohl et al., 2002; Kunz and Kugel, 2015; Nisi et al., 2016; Puskeiler et al., 2016). Insurance claims are limited to inhabited areas where there is sufficient coverage (Kunz and Kugel, 2015; Punge and Kunz, 2016) and they are influenced by the asset vulnerability. Damages to vehicles usually occur for hailstones with diameters > 2 cm (Hohl et al., 2002). While blinds can be damaged by hailstones of 2.5 cm (Stucki and Egli, 2007), windows and tiles require

hailstones of 4 cm or more (Púčik et al., 2019). Consequently, hailstorms producing hail smaller than such diameters will not



be documented in insurance claims, while being identified by weather radars, leading to wrong false alarms in the validation statistics (Kunz and Kugel, 2015).

Another important question relates to the current calibration of the POH metric used in Switzerland. It is based on the third-order polynomial fit by Foote et al. (2005) based in turn on the original observations of Waldvogel et al. (1979). Those
observations were made in Switzerland in 1977 using a dense network of hailpads during the Grossversuch IV field campaign (Federer et al., 1986). Since then, no study has been conducted in Switzerland to verify how the probabilities of hail from POH correspond to current observations and if POH has to be recalibrated.

Since May 2015, users of the free MeteoSwiss app can report hail using a dedicated function (Barras et al., 2019). They can choose from a predetermined set of size categories (see Table 1), and their smartphone GPS location and time are used to
locate and timestamp the report. A 2022 market survey showed that the MeteoSwiss app had a 56% penetration rate among the Swiss population with more than 4.5 million downloads (personal communication from MeteoSwiss). As of October 15, 2023, more than 250000 reports have been collected (Fig. 1) over the Swiss territory (approximately 40000 $km^2$), making it a particularly rich and dense ground-based hail observation dataset.

Crowdsourced observations can contain wrong reports, both intended (jokes) or unintended and must be filtered to keep only
plausible hail occurrences. This is done by identifying suspicious reporting patterns and by checking that a convective cell was present in the neighborhood of the report. In a previous study by Barras et al. (2019), convective cell environments were identified requiring a minimum radar reflectivity of 35 dBZ. However, this filtering method renders the observations dependent on the same radar signal used to compute the hail metrics to be verified. Therefore, we test a spatio-temporal clustering method (ST-DBSCAN; Birant and Kut, 2007) based solely on the data to remove implausible reports.

The aims of this paper are i) to filter crowdsourced hail observations and ii) to make a detailed verification of POH and MESHS and to suggest a potential recalibration of POH. More specifically, we address the following questions:

- What are the advantages and limitations of the spatio-temporal clustering method to filter crowdsourced hail observations?

- What is the skill of POH and MESHS in terms of the hit rate (H), false alarm rate (FAR), Critical Success Index (CSI)
and Heidke Skill Score (HSS)?

- What is the sensitivity of H, FAR, CSI and HSS for POH and MESHS to the distance buffer and to the filtering method?

- Is POH adequately calibrated to be used as a probability of hail at the ground?

- How could POH be recalibrated based on the filtered crowdsourced reports?

We introduce the radar and crowdsourced data in sections 2.1 and 2.2. We present our approach to minimize wrong false
alarms in section 2.3 and our choice of verification period in section 2.4. We discuss the filtering of the crowdsourced data in 2.5 and illustrate the spatio-temporal clustering method in section 2.5.1. We present our approaches for verification in section 2.6. The results of the verification of POH and MESHS with the maximum upscaling approach are presented and discussed in





sections 3.1 and 3.2, respectively. The verification of POH as a probability is presented and discussed in section 3.3 and the recalibration of POH is suggested in section 3.4. Finally, general conclusions and further developments are discussed in section 4. Appendix B follows the first study using crowdsourced reports in Switzerland (Barras et al., 2019) to estimate the fraction of matches with POH and MESHS and makes a comparison with their results. A short analysis of the fraction of matches of POH with observations from automatic hail sensors (Kopp et al., 2023) is presented in appendix C.

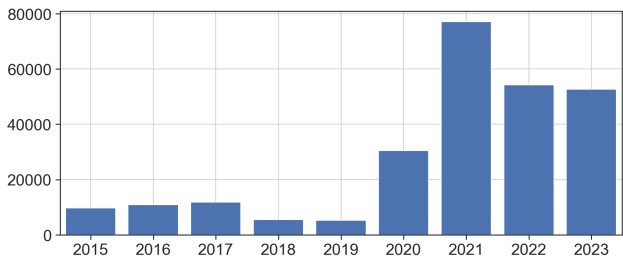

**Figure 1.** Yearly number of hail crowdsourced reports from the MeteoSwiss app.



## 2 Data and Methods

### 2.1 Radar data and radar-based hail metrics

The Swiss weather radar network is composed of 5 identical, dual-polarisation, C-band Doppler radars: 2 on the northern side of the Alps mountain ridge, 2 inside the Alps, and 1 on their southern side (see Fig. 1 of Germann et al. (2022)). All radars are on mountain tops but shielding of the radar beam can occur below 3 km height. This is not a major issue for hail detection and sizing as the relevant altitude range over which the corresponding radar metrics are computed is mostly higher (for more details see Nisi et al., 2016). The antenna scan program has a high resolution in time (5 minutes) and in the vertical direction

(20 elevation angles), which is typically more than in other countries (for more details see Germann et al., 2022). This large number of elevation angles makes Switzerland a unique region to verify POH and MESHS which are based on the vertical structure of radar echoes. The highest altitudes at which 45 and 50 dBZ reflectivity are measured are called echo tops 45 and 50 ($ET45$ and $ET50$). The environmental freezing level height ($H_0$) is retrieved from the Consortium for Small-Scale Modeling numerical weather prediction model (COSMO-1E COSMO, 2021).

The POH metric implemented at MeteoSwiss (Trefalt et al., 2022) is the third-order polynomial fit developed by Foote et al. (2005) based on the Waldvogel et al. (1979) data:

$$y = -1.20231 + 1.00184(ET45 - H_0) - 0.17018(ET45 - H_0)^2 + 0.01086(ET45 - H_0)^3 \tag{1}$$

with $ET45$ and $H_0$ in kilometers. Figure 2 shows the polynomial fit and the corresponding step function. For example, a value of $ET45 - H_0 = 4.2\,km$ corresponds to POH = 80%. POH = 0% for $ET45 - H_0 < 1.65\,km$ and POH = 100% for

$ET45 - H_0 \geq 5.8\,km$.

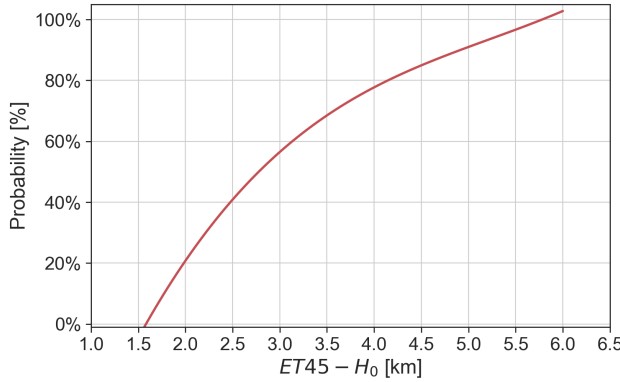

**Figure 2.** Polynomial fit by (Foote et al., 2005) used for POH.



The MESHS metric implemented at MeteoSwiss (Trefalt et al., 2022) follows the nomogram from Joe et al. (2004) based on Treloar (1998). It estimates the maximum hail size on the ground based on the values of $ET50$ and $H_0$ according to the set of equations in chapter 3.1.4 of Trefalt et al. (2022).

Figure 3 shows the corresponding functions for 2, 4, and 6 cm hail size as a function of $ET50$ and $H_0$ in meters.

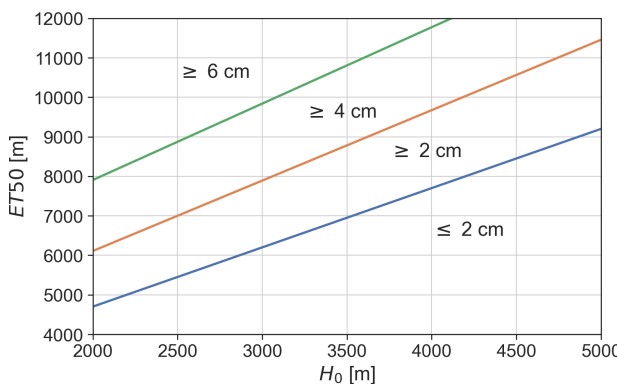

**Figure 3.** Functions upon which the MESHS algorithm is based for 2, 4, and 6 cm maximum hail size. MESHS depends on the relation between $H_0$ [m] on the x-axis and $ET50$ [m] on the y-axis.

For more information about the implementation of POH and MESHS at MeteoSwiss, the reader is referred to Trefalt et al. (2022).

The maximum column reflectivity (CZC) is defined as the largest reflectivity measured over all elevation angles at a given location. CZC is used as a filter to determine if crowdsourced reports were sent in the neighborhood of a convective environment. All radar metrics are defined on a Cartesian grid of 1 $km^2$ cells.





**Table 1.** Successive crowdsourced hail report size category schemes.

| Initial scheme (05.2015 - 09.2017) | Current scheme (old labels) (09.2017-10.2021) | Current scheme (current labels) (11.2021-today) |
| --- | --- | --- |
| No hail | No hail | No hail |
|  | Smaller than coffee bean | < 1 cm (smaller than coffee bean) |
| Coffee bean | Coffee bean | 1 cm (coffee bean) |
| One franc coin | One franc coin | 2 cm (One franc coin) |
| Five franc coin | Five franc coin | 3 cm (Five franc coin) |
| Larger than five franc coin | Golf ball | 5 cm (Golf ball) |
|  | Tennis ball | > 7 cm (Tennis ball) |

## 2.2 Crowdsourced data

The crowdsourcing function of the MeteoSwiss app was introduced in May 2015, and allows users to report the hail size category, time, and location using their smartphone. Each size category is labeled with a reference object. The function was introduced in May 2015 with an initial category scheme, which was extended in September 2017. Then, during October 2021, an explicit size in centimeters was added to the category labels (see Table 1). Before that, the size range corresponding to each category was not explicitly mentioned to the user. The purpose of the category "smaller than coffee bean" was to specifically identify other types of hydrometeors smaller than hail according to its WMO definition (< 0.5 cm World Meterological Association, 2017), such as graupel or sleet. The explicit addition of the size label "< 1 cm" to this category introduced an uncertainty on the type of hydrometeor reported by the user after October, 2021. For this reason, we do not exclude this category to remove potential graupel and sleet observations but rather focus on a specific period of the year (see section 2.4).



 ## 2.3 Approaches for minimizing wrong false alarms

"Wrong false alarms" happen at locations where the radar products indicate the presence of hail and hail reached the ground but no crowdsourced reports are submitted because no one is around to report. To minimize the number of artificial false alarms, we restrict our verification to (i) densely populated regions, to (ii) the time period where the reporting function was easily accessible (2020 onward), and (iii) to the daylight hours when users are awake.

The densely populated areas are derived from the Swiss Federal Statistical Office population estimation for 2021 (STAT-POP21, OFS GEOSTAT). First, the initial population numbers per 100m$^2$ are aggregated and summed on the same Cartesian grid as the radar metrics (1 km by 1 km) and only grid cells with more than 100 inhabitants are retained. A morphological dilation operation using a 3 km × 3 km structuring element is then applied to each grid cell to remove small holes in the densely populated areas and make the region as continuous as possible. The dilation also allows for neighboring pixels to be 140  included, accounting for the potential wind drift of hail. Finally, only continuous areas of 200 $km^2$ or more are retained, to remove small isolated areas mainly located in valleys. The resulting region is called Swiss100 and encompasses 20142 $km^2$ (Fig. 4, green area). We also consider an extended urban area of 1000 $km^2$ centered over Zurich, Switzerland's most populated city, including several other smaller cities. This region is called ZRH (red rectangle in Fig. 4).

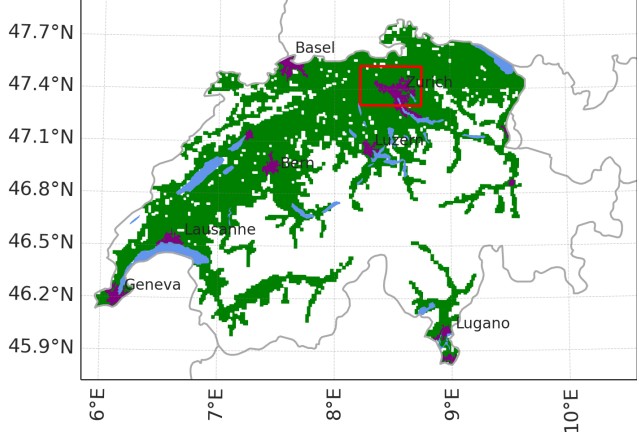

**Figure 4.** Map of Switzerland showing the Swiss100 (green area) and ZRH (red rectangle) regions. The purple patches show the main urban areas of Switzerland according to the Natural Earth populated areas dataset.

    We consider only reports sent under the current category scheme (see Table 1) to avoid the uncertainty of reconciling data 145  from both schemes. From September 2017 to July 2020, the reporting function was more difficult to find to test if it would reduce the number of fake or joke reports (Barras et al., 2019). However, while the report quality improved, this also led to a significant reduction in the number of reports during those years (see Fig. 1). Consequently, our verification uses data from 01 August 2020 to 15 October 2023, to ensure a sufficient report density for the clustering algorithm to work.

    Finally, we consider only grid cells where the peak of the hail (defined as the daily maximum of the radar metric) occurred 150  between 6:00:00 UTC and 21:00:00 UTC to ensure enough users are awake to make reports. If the daily maximum is not



reached in this interval, the corresponding value for that grid cell is set to 0, as if no hail was predicted. Similarly, only reports made during daylight hours, from 6:00:00 UTC to 21:15:00 UTC, are considered (we allow a 15-minute delay for the user to send a report).

Applying those restrictions, we are left with 157795 reports for the Swiss100 region and 18500 for the ZRH area (first two lines of Table 2).





## 2.4 Choice of verification period

The hail or convective season in Switzerland is usually defined as the summer half-year from April to September (Nisi et al., 2016; NCCS, 2021). We found that the season with the largest number of reports without a corresponding POH signal (misses) was spring, with 8 days among the 10 days with the largest number of misses over the Swiss100 region occurring in March,

April, or May. Since October 2021, users have had the option to upload a picture with their report. A visual inspection of some of the pictures sent by users on the spring days with many misses revealed that most of the reports corresponded to hail $< 5mm$, sleet, or graupel. This is coherent with the environmental conditions prevailing in spring that do not allow for deep convective storms that usually produce hail (Doswell, 2001).

     POH and MESHS were originally designed and calibrated using only hail cases from the summer months (June, July and

August): Waldvogel et al. (1979) explicitly discarded cases of graupel/sleet showers originating from cold lows while the hail observations used by Treloar (1998) have diameters $\geq 5mm$. Consequently, we decided to focus on the summer season (June to August) to remove the hail $< 5mm$ and sleet cases happening in spring and make the verification in conditions similar to their calibration. Nevertheless, we note that the number of misses (and hence the hit rate) would be slightly lower if one considers the entire convective season. The remaining reports are 106923 for the Swiss100 region and 10941 for the ZRH region (Table

2).





## 2.5 Filtering methods for the crowdsourced reports

First, the data is preprocessed according to the following filters: any duplicate of the same user ID, time (rounded to 5 minutes), coordinates (rounded to 1 km), and size is removed. Suspicious reporting patterns are identified according to the filters described in Barras et al. (2019) with an additional filter identifying users sending more than 4 reports per day. The corresponding reports are removed. We also apply the time filter to discard reports with more than 30 minutes difference between the submission time and event time (in case the user manually changed the time). Finally, we compare two distinct methods to check the plausibility of the remaining preprocessed reports:

- The reflectivity filter from Barras et al. (2019): only reports with CZC $\geq$ 35 dBZ within 4km radius and 15min time are kept (B19 filter)

- A new spatio-temporal clustering approach (ST-DBSCAN Birant and Kut, 2007) proposed in this paper (HRC filter)

ST-DBSCAN was first introduced by Birant and Kut (2007) as an extension of the existing DBSCAN algorithm (Ester et al., 1996) to data with time dimension. DBSCAN stands for Density-Based Spatial Clustering of Applications with Noise, it can discover clusters of data with arbitrary shapes and does not require predetermining the number of clusters. We use the Python implementation of the algorithm presented by Cakmak et al. (2021) as the basis for the hail report clustering (HRC) filter. The principle of the HRC filter is that if two reports are within a given distance (EpsD) and time window (EpsT), they are grouped together. We require at least 5 grouped reports to form a cluster, and only clustered reports are retained and the others are removed. The idea behind keeping only clustered reports is that if several users located in the same spatial neighborhood send hail observations within a short time window then the plausibility that hail occurred is increased compared to single, isolated reports. We illustrate the HRC filter and discuss the choice of the parameters in the following section.





### 2.5.1 Illustration of the HRC and B19 filters: example of June 20, 2021

Figure 5a shows the hail observations of 20 June 2021. This day featured widespread and intense hail activity with several storms crossing the northern part of Switzerland from southwest to northeast. An extended region of large MESHS values (red areas) enclosed by the daily maximum POH contour (in green) is visible. 4164 crowdsourced reports were sent and 5 automatic hail sensors (Appendix C) recorded a total of 120 impacts. Figure 5b shows when the crowdsourced reports were sent. Groups of reports sent during the same UTC hour have the same color, and reports sent before or after stand out with their different color. Such time outliers can be intended false reports (jokes) or non-intended errors made by users. Non-intended errors can be made when a user witnesses hail, and cannot report at the moment but does so several hours later and forgets to change the report's time (or location) manually.

Figure 6a shows the results of the application of the clustering algorithm with a distance parameter (EpsD) of 12 km and a time parameter (EpsT) of 12 minutes. The clustering algorithm creates groups of reports similar to those identified in Fig. 5b (colored dots) and removes the time and distance outliers (grey dots). We can apply the B19 filter to the same data (Fig. 6)b and compare the results of the two filters. Most reports are either retained (green dots, 3321 reports) or removed (grey dots, 557 reports) by both filters. 124 reports are retained by the HRC filter but removed by the B19 filter. As it is extremely unlikely that hail occurs below 35 dBZ, those reports are likely intended false reports or non-intended errors. This is one limitation of the HRC filter: it can retain clusters composed of false reports and errors, or, more likely, retain false reports or errors accidentally clustered with correct reports. However, 124 reports represent a small fraction ( 3%) of the total reports. Finally, 162 reports are removed by the HRC method while retained by the B19 method.

### 2.5.2 Parameters of the HRC filter

Whether a report is included in a cluster depends on the EpsD and EpsT parameters. We consider the 3 following sets of parameters to assess the sensitivity of our results to the choice of parameters:

- 8 km distance and 8 minutes maximum time (8km/8min)

- 12 km distance and 12 minutes maximum time (12km/12min)

- 16 km distance and 16 minutes maximum time (16km/16min)

The pairs of distance and time correspond to a 60 $km\,h^{-1}$ propagation velocity of the storm, allowing the capture of fast-moving hailstorms in Switzerland (Nisi et al., 2018). They also account for the fact that users may not send their reports immediately when they see hail. We also looked at shorter distances and times but found that they were too small to capture all the relevant reports. In contrast, larger distances and times resulted in clustering unrelated reports.

### 2.5.3 Motivations for using the HRC and B19 filters

The B19 filter efficiently removes reports sent without a storm occurring at the report location and time. However, it is not fully independent from the POH and MESHS products which use the 45 dBZ and 50 dBZ echotops, respectively. On the other hand,





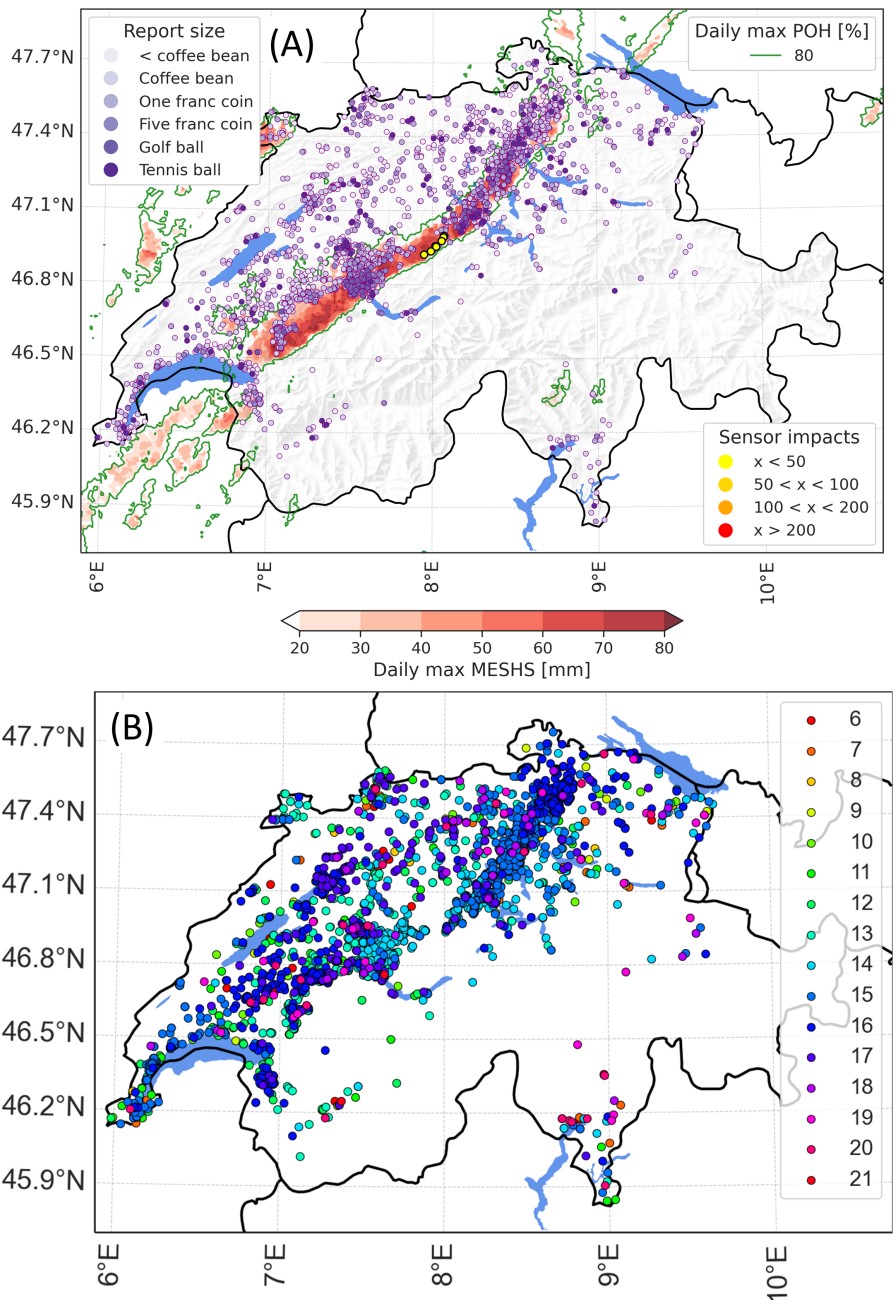

**Figure 5.** (a) Hail observations of 20 June 2021: POH≥80% areas (green contours), daily maximum MESHS (red color scale), location of crowdsourced reports (purple dots, largest sizes are darker), sensor impacts (yellow to red dots). (b) 4164 crowdsourced reports of 20 June 2021, colored according to the UTC hour at which they were sent.





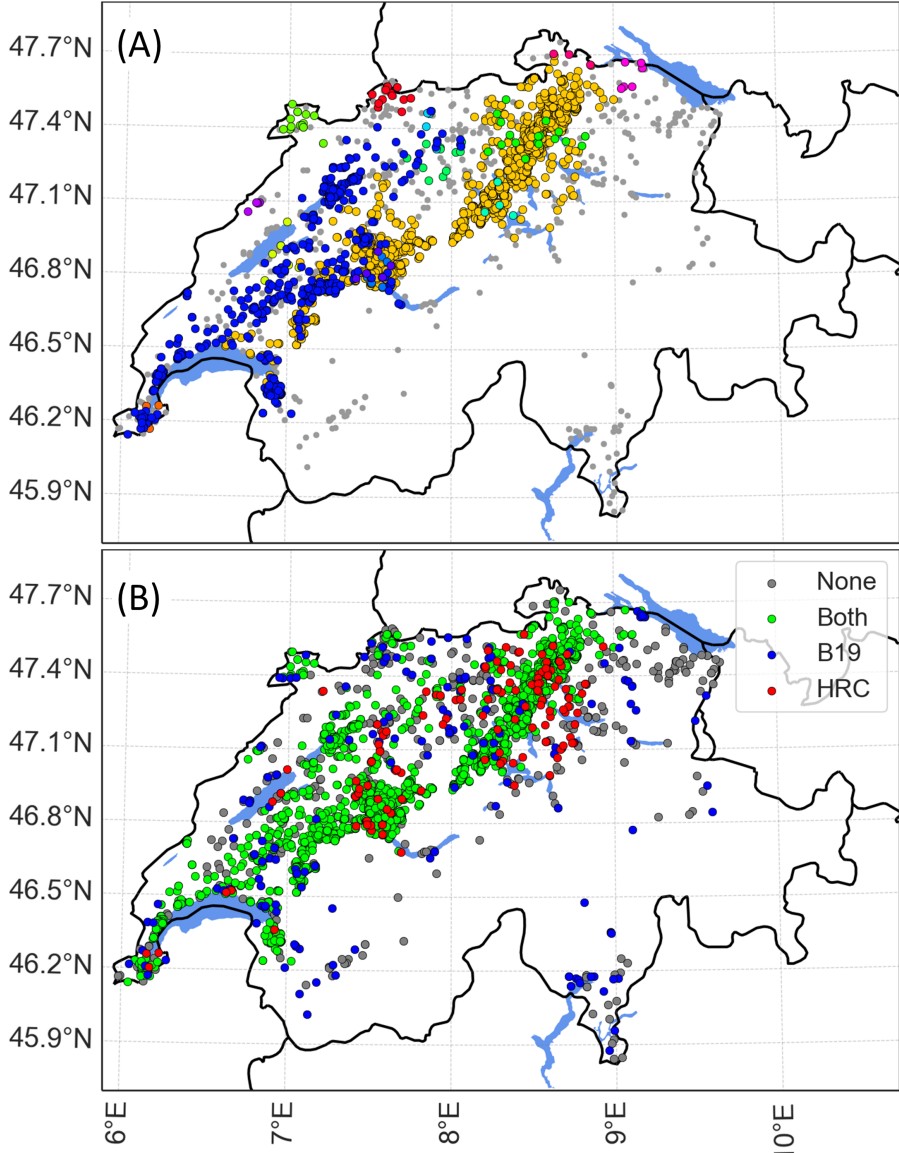

**Figure 6.** (a) 4164 crowdsourced reports of 20 June 2021. Colored dots show clustered reports (EpsD = 12km and EpsT = 12min) and grey dots show reports that are not part of a cluster. (b) 4164 crowdsourced reports of 20 June 2021. Green dots show the reports retained by both the HRC and the B19 filter; Red dots show the reports only retained by the HRC filter; Blue dots show the reports only retained by the B19 filter; Grey dots show the reports removed by both filters.

the HRC filter relies solely on the data itself, making it fully independent from the verified radar metrics. The main limitation of the HRC filter is that it assumes that the app coverage and the population density are sufficient to generate enough reports to



be clustered. The precautions taken to minimize wrong false alarms described in section 2.3 also help ensure that those criteria are met.

If we apply the B19 and HRC filters to the complete set of reports (first line of table 2), we find that they agree on 87% (EpsD = 8 km, EpsT = 8 minutes) to 96% (EpsD = 16 km, EpsT = 16 minutes) of the reports (i.e. the report is either retained or removed by both filters. Both filters can also be combined by clustering only reports retained by the B19 filter, or by first applying the HRC filter and then the B19 filter on the retained clustered reports. We also test the latter combination (B19 + HRC). Such a combination is not fully independent from the radar metrics. However, it removes part of the false reports and

errors that may have been clustered with correct reports, further improving the quality of the observations.



## 2.6 Verification framework

We verify the hail radar metrics against the hail observations from the crowdsourced reports using a maximum upscaling approach (see eg. Ebert, 2008) to incorporate the distance buffer. We start by considering each day of the period and compute a hail radar detection grid and a hail observations grid. Cells of the radar detection grid where the daily maximum value of the radar metric is equal to or above a certain threshold are set to 1 (1-cell), and the other cells are set to 0 (0-cell). For POH, threshold values ranging from 1% to 100% with intermediate steps of 10% are considered, whereas MESHS values of 2, 3, 4, and 6 cm are considered. Cells of the observations grid with at least one filtered crowdsourced report are set to 1, and the other cells are set to 0. Days without at least one grid cell of the region with either a hail observation or a hail radar detection are discarded.

Hail radar detection and observation grids are computed on consecutively larger squared areas of $1 \times 1$, $2 \times 2$, up to $10 \times 10$ grid cells (1, 4, up to 100 $km^2$, respectively) by taking the maximum over that area, the $1 \times 1$ area simply being a grid cell by grid cell comparison. An area with radar detection confirmed by an observation is classified as a hit (A). An area with radar detection not confirmed by an observation is classified as a false alarm (B). An area without radar detection but with an observation is classified as a miss (C). Finally, an area without radar detection and an observation is considered a correct negative (D). The second row of Fig. 7 illustrates this approach using a $3 \times 3$ grid cells area on two initial situations.

From these outcomes, the hit rate (H), false alarm ratio (FAR), critical success index (CSI) and Heidke Skill Score (HSS) are calculated as follows (Hogan and Mason, 2011):

$$H = \frac{A}{A+C} \qquad\qquad FAR = \frac{B}{A+B} \qquad\qquad CSI = \frac{A}{A+B+C}$$

$$HSS = \frac{2 * (A * D - B * C)}{(A+C) * (C+D) + (A+B) * (B+D)}$$

The range of H, FAR, CSI is from 0 to 1, with 1 for a perfect detection for H and CSI, and 0 for FAR. The range of HSS is $-\inf$ to 1, with 1 for a perfect detection. H is sensitive to hits but ignores false alarms, increasing with overdetecting events. FAR is sensitive to false alarms but ignores misses, increasing with underdetecting events. CSI measures the fraction of correct detections and includes false alarms and misses. HSS measures the fraction of correct detections after eliminating those detections that would be correct due purely to random chance. It includes false alarms and misses. The maximum upscaling approach hence helps us to verify the performance and skill of the radar metric.

We use a different approach to verify that POH is calibrated as probability. We would like to answer the question: What is the probability of observing hail given that a POH signal detects hail within a certain distance? Holleman (2001) suggested that a hail detection method (the Waldvogel criteria or any other) can be used to produce a probability of hail defined as POH = 1 - FAR, where FAR is the false alarm ratio as defined above. A radar grid cell value above a certain threshold indicates a probability of hail which is equal to 1 - FAR at that threshold.



The FAR values calculated using the maximum upscaling approach could also be used to compute the probability of observing hail. However, the results would depend on the shape of the upscaling area and on how the shape is applied to the observation and radar grids (border effects). We prefer to use a quantity that depends only on the relative position of the observations and radar detections, which we call $FAR_{prob}$ to avoid any confusion with the FAR computed with the upscaling approach. $FAR_{prob}$ is computed similarly to the fraction of matches described in appendix B. Each 1-cell of the radar grid is considered a hit if the distance between the center of the cell and a report location is within a matching distance. Otherwise, it is a false alarm. $FAR_{prob}$ is simply the number of false alarms divided by the total number of 1-cell of the radar grid. The third row of Fig. 7 illustrates this approach using a 1 km matching distance. $FAR_{prob}$ cannot be used to verify the performance and skill of the radar metric because a single observation cell can be used to match several radar cells.

Finally, two sets of crowdsourced reports are considered for the verification: one with all the size categories (complete set) to verify POH and another with categories $> 2\,cm$ (1 CHF coin, 5 CHF coin, Golf ball and Tennis ball; hereafter the $> 2\,cm$ set) for verifying MESHS. For this second set, the clustering algorithm is performed only with the four selected categories. Table 2 shows the corresponding number of reports by region, filtering method and set.





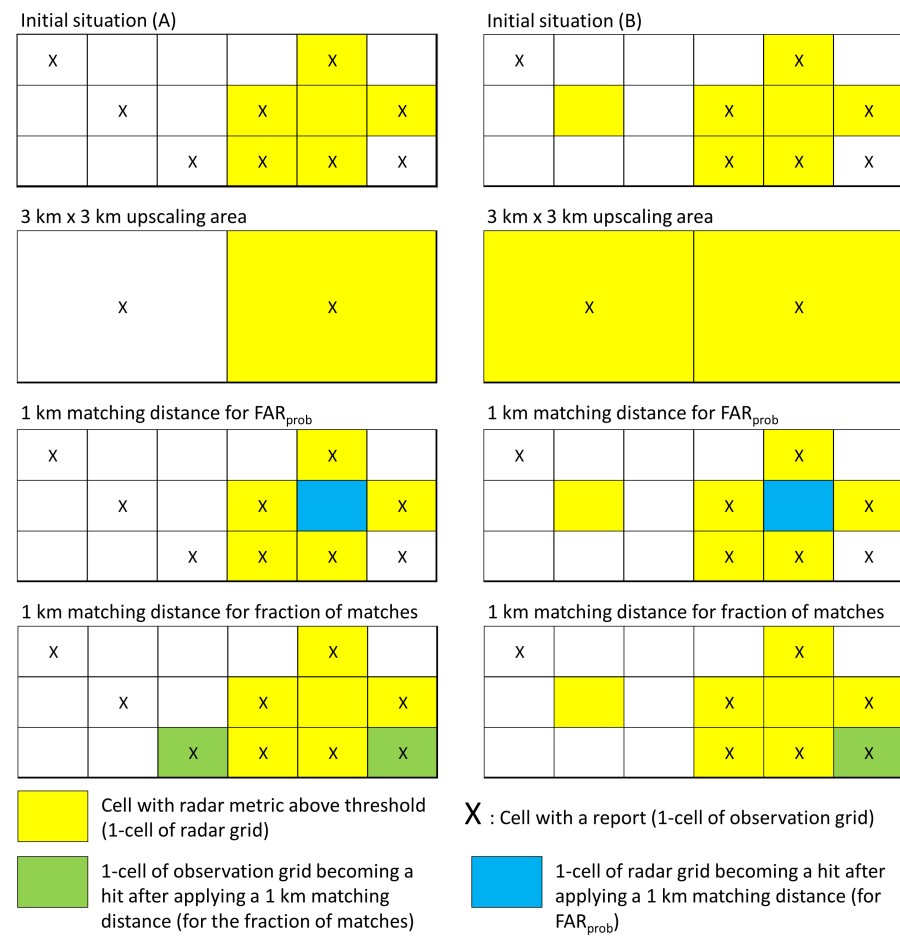

**Figure 7.** Illustrations of the maximum upscaling approach (second row) and of the calculation of $FAR_{prob}$ (third row) and the fraction of matches (bottom row, see appendix B) for two initial situations (A) and (B). In (A): H = 0.56, FAR = 0.17 initially; H = 0.5, FAR = 0 after applying a 3 km × 3 km maximum upscaling; $FAR_{prob}$ = 0 and fraction of matches = 78% for a 1 km matching distance. In (B): H = 0.71, FAR = 0.29 initially; H = 1, FAR = 0 after applying a 3 km × 3 km maximum upscaling; $FAR_{prob}$ = 0.14 and fraction of matches = 86% for a 1 km matching distance.



**Table 2.** Number of crowdsourced reports per region, season and filtering method from 01.08.2020 to 15.10.2023. In parenthesis: reports with size $> 2\,cm$.

| Region | Season | Filtering method | Observations number Complete set ($> 2cm$ set) |
|---|---|---|---|
| Swiss100 | Year | None | 157795 (42904) |
| ZRH | Year | None | 18500 (3908) |
| Swiss100 | Summer | None | 106923 (35147) |
| ZRH | Summer | None | 10941 (2928) |
| Swiss100 | summer | HRC 8km/8min | 80757 (23707) |
| Swiss100 | summer | HRC 12km/12min | 84792 (24938) |
| Swiss100 | summer | HRC 16km/16min | 87804 (25944) |
| Swiss100 | summer | B19 | 92467 (29019) |
| Swiss100 | summer | B19 + HRC 8km/8min | 80499 (23568) |
| Swiss100 | summer | B19 + HRC 12km/12min | 83945 (24673) |
| Swiss100 | summer | B19 + HRC 16km/16min | 86047 (25474) |
| ZRH | summer | HRC 8km/8min | 8302 (1726) |
| ZRH | summer | HRC 12km/12min | 8616 (1812) |
| ZRH | summer | HRC 16km/16min | 8876 (1885) |
| ZRH | summer | B19 | 9213 (2212) |
| ZRH | summer | B19 + HRC 8km/8min | 8283 (1722) |
| ZRH | summer | B19 + HRC 12km/12min | 8516 (1799) |
| ZRH | summer | B19 + HRC 16km/16min | 8692 (1853) |





# 3 Results and Discussion

## 3.1 Verification of POH with the upscaling approach

In this section, we use the maximum upscaling approach to analyze the skill of POH over the ZRH region in terms of the hit rate (H), False alarm ratio (FAR), Critical Success Index (CSI), and Heidke Skill Score (HSS), as a function of the POH threshold. The complete set of observations is used (all report sizes). We assess the sensitivity of the results to the upscaling area and to the filtering method and compare our results with previous studies. The results for the Swiss100 region are shown in appendix A.

Figure 8 shows the results using the B19 + HRC 8km/8min filter for various upscaling areas, an upscaling area of 1km x 1km being equivalent to the original grid resolution. The FAR, CSI and HSS improve when increasing the upscaling area, while H remains almost constant. The fact that H does not improve more significantly with increasing upscaling area might seem counterintuitive at first sight, but it is a consequence of the patterns of the observations and detections grids. For example, applying a 3 km × 3 km upscaling on the situation (A) of Fig. 7 results in a decrease of H from 0.56 to 0.5, while the same upscaling on the situation (B) of Fig. 7 results in an increase of H from 0.71 to 1. The largest improvement occurs when passing from the original grid cell resolution to a 2 km × 2 km area, and going from a 5 km × 5 km to a 10 km × 10 km results only in minor improvement.

Previous studies (Hohl et al., 2002; Schuster et al., 2006) found that the wind drift effect on hail can reach between 2 km and 3 km. As mentioned in the introduction, most previous studies implicitly incorporated this wind drift effect because the spatial resolution of their observations (or radar detections) was coarser (10km or more). However, a wind drift larger than 5 km seems physically unlikely even in strong wind conditions, and we will further discuss and compare the results obtained with a 4 km × 4 km area.

Figure 9 compares the results of the different filtering methods considering a 4 km × 4 km area. For the HRC methods, H and, to a lesser extent, FAR increase with decreasing space and time parameters. We have seen that smaller distance and time parameters are more stringent conditions for reports to be clustered, thus reducing the number of hail observations for verification. With fewer observations, false alarms from the radar metrics are more likely to occur. The higher H corresponding to smaller distance and time parameters might be explained by the fact that clustering effectively removes spatial outliers corresponding to (wrong) observations made far from a hailstorm. We see that applying the B19 filter on top of the HRC filter (B19 + HRC) has a very limited impact on all scores. In fact, the HRC 8km/8min (red) and B19 + HRC 8km/8min (pink) curves are superimposed for all scores (Fig. 9).

The CSI and HSS values are very close for the six methods incorporating HRC (Fig. 9), with the B19 + HRC 8km/8min reaching the highest values for the CSI (0.37) and the HSS (0.52) at a threshold of 60%. For the Swiss100 region CSI and HSS peak at 80% (see appendix A). It is interesting to note that the optimal threshold for the Swiss100 region according to CSI and HSS corresponds to the one that has often been used in the literature to derive hail days (Nisi et al., 2016, 2018; Madonna et al., 2018; NCCS, 2021). We also note that both CSI and HSS for the ZRH region are almost constant between thresholds of 1% to 80%, such that the 80% is also close to optimal for this region.



The B19 method (blue curve in Fig. 9) has the lowest (worst) H and the lowest (best) FAR. This method filters out fewer reports than the HRC methods, explaining its low FAR, and potentially keeps more wrong reports, explaining the low H. The resulting CSI and HSS values are visibly lower than those obtained when incorporating the HRC filtering.

       The FAR values for a 4km x 4km area are between 0.3 (B19 filter and 100% threshold) and 0.7 (B19 + HRC 8km/8min and 1% threshold), reaching 0.45-0.55 for the best corresponding HSS values (0.48-0.52, Fig. 9). This is lower than in some
previous studies verifying POH despite their use of a coarser grid resolution. Kunz and Kugel (2015) found values between 0.7 and 0.8 for a 10 km grid cell, using hail damage claims to buildings in Baden-Württemberg, Germany. Nisi et al. (2016) found values between 0.49 and 0.95 for 25 urban areas, each being 10's of $km^2$ wide, using car insurance reports. Such higher FAR values are likely explained by the fact that buildings and cars are damaged by hailstones at least larger than 2 cm. In contrast, crowdsourced reports capture all hail sizes and have better area coverage.

Holleman (2001) found slightly lower FAR values (0.2 to 0.5) using observations from weather stations and damage reports from agricultural insurance companies. However, they accounted for the fraction of unreported events in their approach and allowed for a positioning tolerance of 12.5 km. Puskeiler et al. (2016) found slightly lower FAR values (0.2 to 0.6) and higher HSS values reaching up to 0.7, for their dataset of hail damage claims to buildings, and 5 km × 5 km area.

       We also note that Waldvogel et al. (1979) reported a FAR of 0.5 using data from hailpads, and considering individual storms.
Kessinger et al. (1995) verified four hail detection algorithms, including a version of POH, and found FAR values smaller than 0.06 for all of them (0.04 for POH). Such values are surprisingly low compared to the rest of the literature and might be due to their consideration of hydrometeors < 5 mm as hail, their 15 km influence region, or their storm selection process, which is not detailed in the above-mentioned reference.



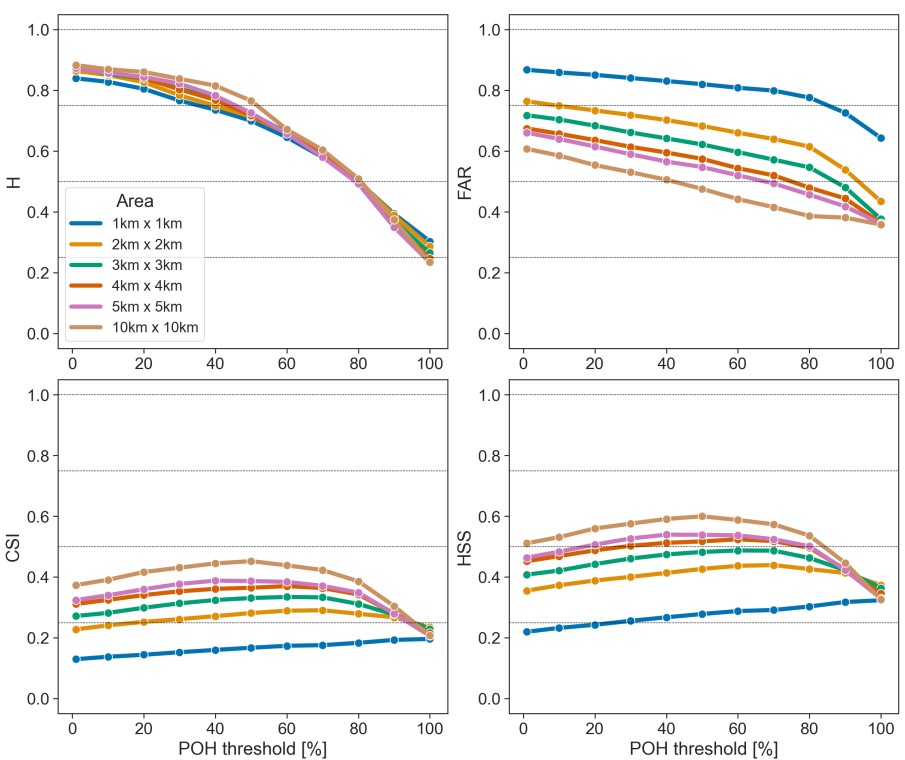

**Figure 8.** POH hit rate (H, top left panel) and False alarm ratio (FAR, top right panel), Critical Success Index (CSI, bottom left) and Heidke Skill Score (HSS, bottom right) for the ZRH region, using the B19 + HRC 8km/8min filter applied on the complete set of observations, stratified by upscaling area (colored curves).




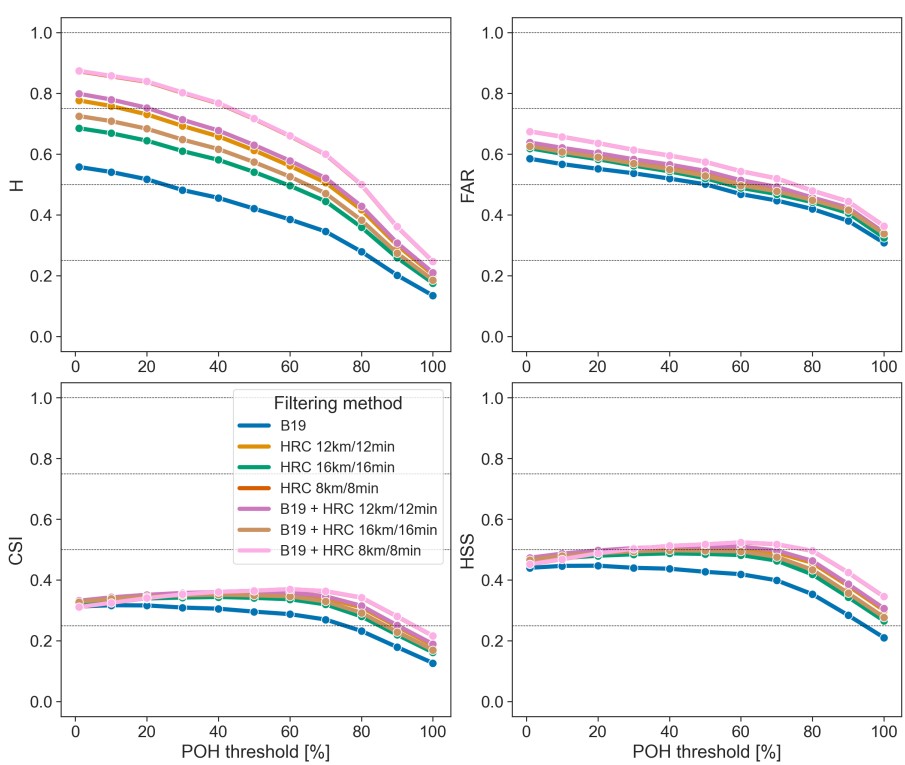

**Figure 9.** POH hit rate (H, top left panel) and False alarm ratio (FAR, top right panel), Critical Success Index (CSI, bottom left) and Heidke Skill Score (HSS, bottom right) for the ZRH region, using the complete set of observations and an upscaling area of 4 km × 4 km, stratified by filtering methods (colored curves).



## 3.2 Verification of MESHS with the upscaling approach

We proceed similarly to the previous section to analyze the skill of MESHS. We use the $> 2cm$ set of observations and keep
the size categories equal to or larger than the verified MESHS threshold: 1 CHF, 5 CHF, golf ball and tennis ball for MESHS
2 cm; 5 CHF, golf ball, and tennis ball for 3 cm; golf ball and tennis ball for 4 cm; and tennis ball for 6 cm. We mention here
that the reports of the tennis ball size category might be less reliable as this category supposedly contains most of the joke
reports (Barras et al., 2019). Therefore all results for the MESHS 6 cm threshold should be taken with care. The results for the
Swiss100 region are also shown in appendix A.

Figure 10 shows the results using the B19 + HRC 8km/8min filter for the different upscaling areas. As for POH, the FAR,
CSI and HSS improve when increasing the upscaling area, but the effects on H depend on the MESHS threshold. All scores
worsen with increasing MESHS thresholds, except for the 6 cm threshold of the 10 km × 10 km area where all scores strongly
improve compared to the 4 cm threshold. As mentioned above, the results for 6 cm should be taken with care. We note that this
effect is not visible for the Swiss100 region (see Fig. A3 in Appendix A).

Figure 11 compares the results of the different filtering methods considering a 4 km × 4 km area. As in the case of POH, the
B19 + HRC 8km/8min filter (pink curve) gives the better scores, and the B19 filter alone has the worst scores. The difference
is particularly striking for H. All methods incorporating HRC have similar CSI and HSS. The FAR is comparable between all
methods and remains relatively high (between 0.6 and 0.85). The same high level of false alarms was also reported by Schmid
et al. (2023), who used MESHS to calibrate insurance hail damage impact functions for buildings and cars. To our knowledge,
no other study looked at the FAR of MESHS.

The high FAR of MESHS raises the question of whether MESHS is useful in estimating the maximum hail size. The answer
to this question is not straightforward. First, we note that by definition MESHS is the "Maximum Expected Severe Hail Size"
over a 1 $km^2$ grid cell and that the largest hailstone over this area might just not be observed by the MeteoSwiss app users,
even in densely populated areas. Second, it is not clear if this maximum expected size should always occur, or if this is only
a necessary but not sufficient condition. In other words, does a MESHS value of 4 cm mean that the conditions for producing
a 4 cm hailstone are met, but it won't systematically happen; or does it mean that a hailstone of such a diameter should be
produced in all cases? In the first case, not observing a 4 cm hailstone is not a false alarm, whereas it is in the second. For
MESHS to be useful, its corresponding hailstone size should be observed at least more often than not, which means a FAR
value of 0.5 or lower.

Finally, we also point out that the sample size of the observations gets smaller as the MESHS size threshold increases. For
example, the sample size using the B19 + 8km/8min clustering filter for the ZRH region has 1722 observations for 2 cm, 296
for 3 cm, 75 for 4 cm, and 20 for 6 cm over the ZRH region and 23568, 6607, 2176, and 386 respectively for the Swiss100
region. This is significantly less than the complete set of observations (see Table 2) used for POH but still significantly more
than the sample size used to define MESHS originally (e.g.: 27 observations; Treloar, 1998).



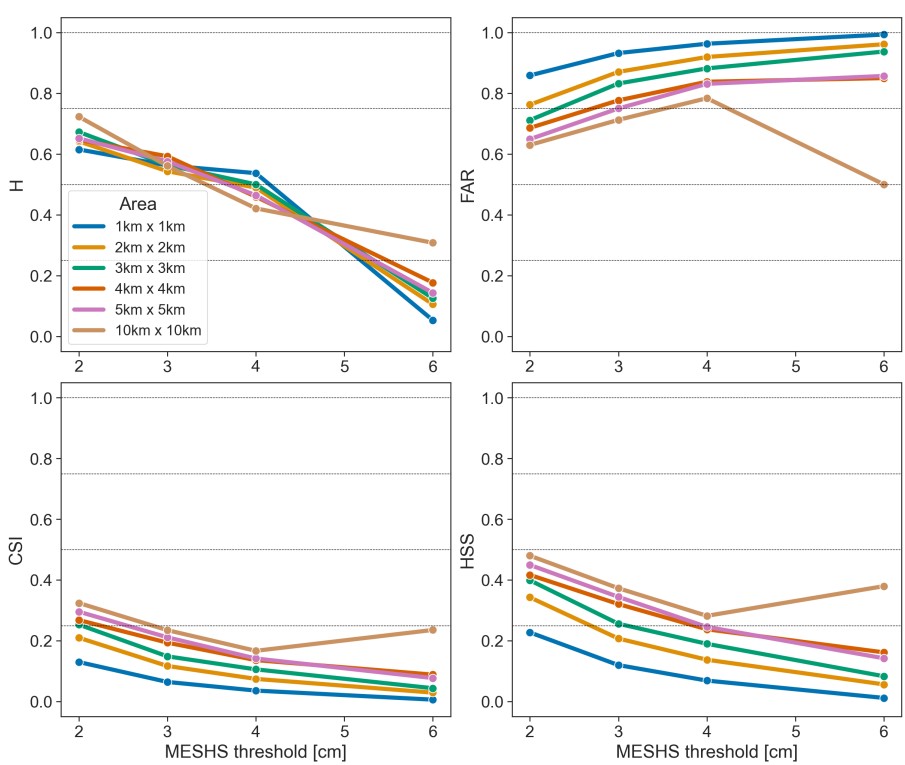

**Figure 10.** MESHS hit rate (H, left panel) and False alarm ratio (FAR, right panel), Critical Success Index (CSI, bottom left) and Heidke Skill Score (HSS, bottom right) for the ZRH region, using the B19 + HRC 8km/8min filter applied on the $> 2cm$ set of observations, stratified by upscaling area (colored curves). Each MESHS size threshold is verified against size categories equal to or larger than the threshold.

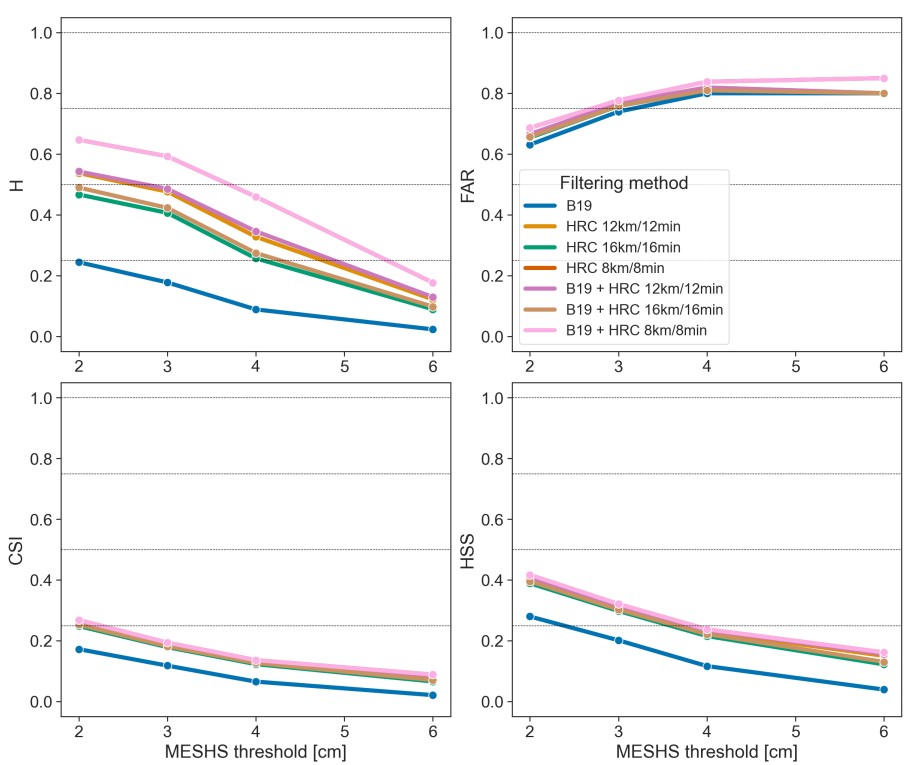

**Figure 11.** MESHS hit rate (H, left panel) and False alarm ratio (FAR, right panel), Critical Success Index (CSI, bottom left) and Heidke Skill Score (HSS, bottom right) for the ZRH region, using the $> 2cm$ set of observations and an upscaling area of 4 km × 4 km, stratified by filtering methods (colored curves). Each MESHS size threshold is verified against size categories equal to or larger than the threshold.



### 3.3 Verification of POH as a probability

In this section, we compute $FAR_{prob}$ as a function of the POH threshold over both the ZRH and Swiss100 region. The complete set of observations is used (all report sizes). We assess the sensitivity of the results to the matching distance and to the filtering method and briefly comment on how they differ from the results obtained with the upscaling approach. $FAR_{prob}$

is analyzed to determine if the current POH calibration is comparable to a probability as suggested by Holleman (2001).

Figure 12 shows the results using the B19 + HRC 8km/8min filter for different matching distances, a matching distance of 0 km (blue curve) corresponding to the original grid resolution (and therefore also to an upscaling area of 1 km × 1 km). $FAR_{prob}$ improves (decreases) with increasing matching distance. The decrease in $FAR_{prob}$ remains significant up to 4 km (and even for larger distances, not shown). This might indicate that POH overestimates the hailswath area, as increasing the

matching distance will convert the false alarm grid cells surrounding a report into hits. This overestimation was also noticed for MESHS by Schmid et al. (2023).

$FAR_{prob}$ is lower for the ZRH region (Fig. 12, right panel) than for the Swiss100 region (Fig. 12, left panel). This is because the ZRH region is more densely populated than the Swiss100 region. Some areas of the Swiss100 region might be less populated or even without population, due to the dilation operation on the original cell selection based on a density of 100

people per $km^2$. With a higher average population density and fewer subareas without population, it is less likely for an actual hailstorm to be missed over the ZRH region. We note here that H and FAR obtained with the upscaling approach are also lower for the ZRH region than for the Swiss100 region (see Fig. 8 in section 3.1 and Fig. A1 in A).

We selected a matching distance of 2 km to account for the wind drift of hail and to further compare the results of the different filtering methods in Fig. 13. The results are consistent with those obtained with the upscaling approach (Fig. 9). The

largest $FAR_{prob}$ is reached by the B19 + HRC 8km/8min filter (pink curve), while the smallest p$FAR_{prob}$ is obtained by the B19 filter alone (blue curve).

The definition $POH = 1 - FAR_{prob}$ works properly if the maximum and minimum value of $FAR_{prob}$ are close to 1 and 0, respectively. Based on the results from Fig. 12 and 13, we see that the range of values covered by the $FAR_{prob}$ curves are too restraint to compute a probability. The curves having a maximum $FAR_{prob}$ close to 0.9 have a minimum of 0.7 and those

having a minimum $FAR_{prob}$ close to 0.2 never get higher than 0.6. Consequently, a recalibration of POH is necessary to have a well-defined probability. This is done in section 3.4.




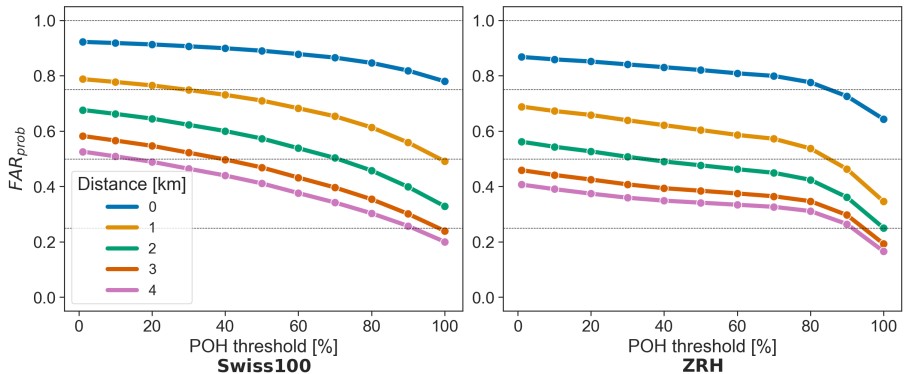

**Figure 12.** POH $FAR_{prob}$ for the Swiss100 (left) and ZRH (right) regions, using the B19 + HRC 8km/8min filter and stratified by matching distance (colored curves).

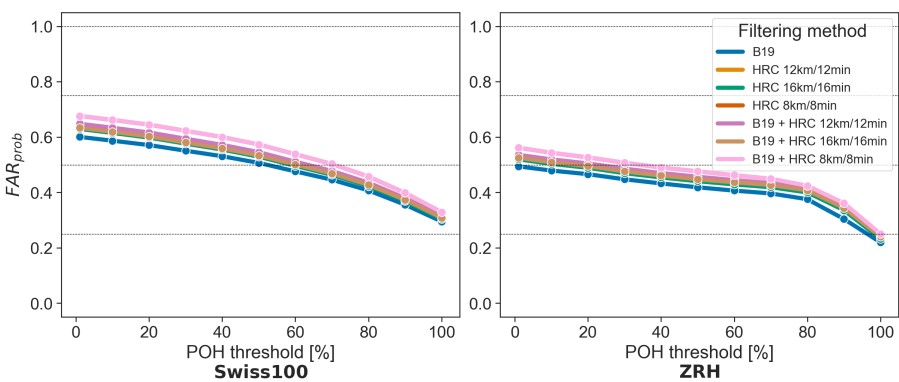

**Figure 13.** POH $FAR_{prob}$ for the Swiss100 (left) and ZRH (right) regions, using a matching distance of 2 km and stratified by filtering methods (colored curves).





## 3.4 Recalibration of POH

The results of section 3.3 suggest that the initial range of $ET45 - H_0$ values used for POH (1.65 to 5.8 km) is not wide enough to compute a probability from the quantity $1 - FAR_{prob}$. In this section, we consider values ranging from -3000 m to 12000 m for $ET45 - H_0$, and compute $1 - FAR_{prob}$ for each interval of 500 m, using a 2 km matching distance. We apply the same matching distance approach as described in section 2.6 except that in this case, $ET45 - H_0$ must be within an interval of x and x + 500 m (with $x = -3000, -2500, -2500, ..., 11500m$) and not above some threshold.

Figure 14 shows the results for the ZRH (red) and Swiss100 (green) regions using a B19 + HRC 8km/8min filtering method, compared to the original POH curve (grey, see 1). The red and green curves are cubic fits based on the data of each region and read:

$$y_{ZRH} = 0.1581 + 0.0876(ET45 - H_0) + 0.0069(ET45 - H_0)^2 - 0.0007(ET45 - H_0)^3 \qquad (2)$$

$$y_{Swiss100} = 0.0603 + 0.0628(ET45 - H_0) + 0.0122(ET45 - H_0)^2 - 0.0098(ET45 - H_0)^3 \qquad (3)$$

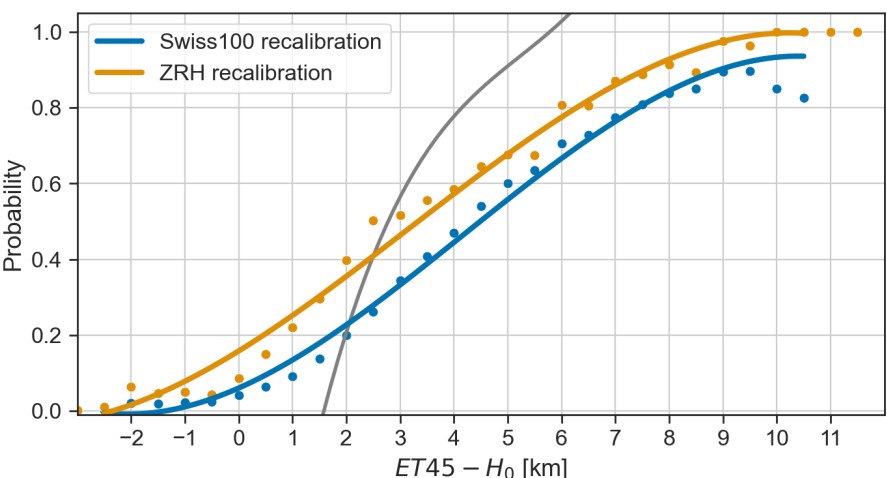

**Figure 14.** Recalibration of POH for the ZRH (red) and Swiss100 (green) regions using a B19 + HRC 8km/8min filtering method and a 2 km matching distance. The grey curve is the original POH calibration.

Compared to the original POH calibration, we see that according to the observed data, non-zero probabilities exist for values of $ET45 - H_0 \leq 1.65\,km$ and that the probability is less than 100% for $ET45 - H_0 = 5.8\,km$. The new fits are less steep than the original one and lead to higher probabilities of hail for $ET45 - H_0 < 2\,km$ and lower probabilities for $ET45 - H_0 > 3\,km$. The use of an extended range of $ET45 - H_0$ values results in curves more consistent with a probability, which flatten around 0 and 1, contrary to the original curve. We note that negative values of $ET45 - H_0$ still correspond to a non-zero probability of hail. Such cases could still be related to users reporting graupel or sleet that may still happen in summer. Another likely





explanation is that (small) hail can still occur at maximum column reflectivity slightly below 45 dBZ. In such cases, there is
no echotop 45 (ET45 = 0) and with a typical freezing level height in summer being above 3000 m, this can lead to the negative
values observed for $ET45 - H_0$.

The probability for the Swiss100 region is lower by approx. 10%-15% than for the ZRH region for the same $ET45 - H_0$
value. This is related to the lower level of false alarms of the ZRH region compared to the Swiss100 region (see Fig. 13) and,
as discussed in section 3.3, most likely related to the higher population density of the ZRH region. The Swiss100 calibration
is likely more robust because of its larger sample size, while the level of false alarms for the ZRH calibration might be more
realistic.

The calibration is relatively stable with respect to the choice of the filtering method, as shown by Fig. 15, but it strongly
depends on the choice of the matching distance as can be seen in Fig. 16. Indeed, the matching distance is an integral part of
the definition of the probability metric. Strictly speaking, Eq. 2 and Eq. 3 give "the probability to observe hail within a 2 km
radius from the center of a grid cell having a given value of $ET45 - H_0$". This probability increases (decreases) with increasing
(decreasing) matching distance which is what Fig. 16 shows. The original definition of POH does not explicitly incorporate
this notion of distance and was likely assumed to be implicitly related to the spatial resolution of the radar grid ($1\,km^2$).

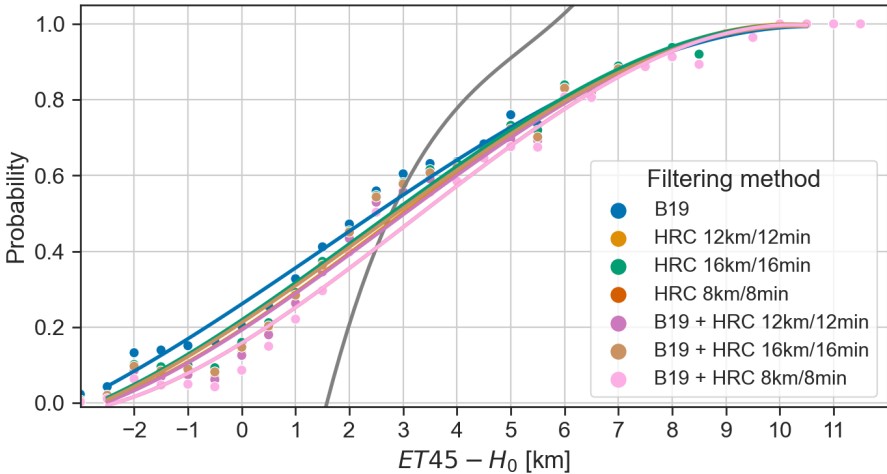

**Figure 15.** Recalibration of POH for the ZRH region using a 2 km matching distance and for the different filtering methods (colored curves).
The grey curve is the original POH calibration.



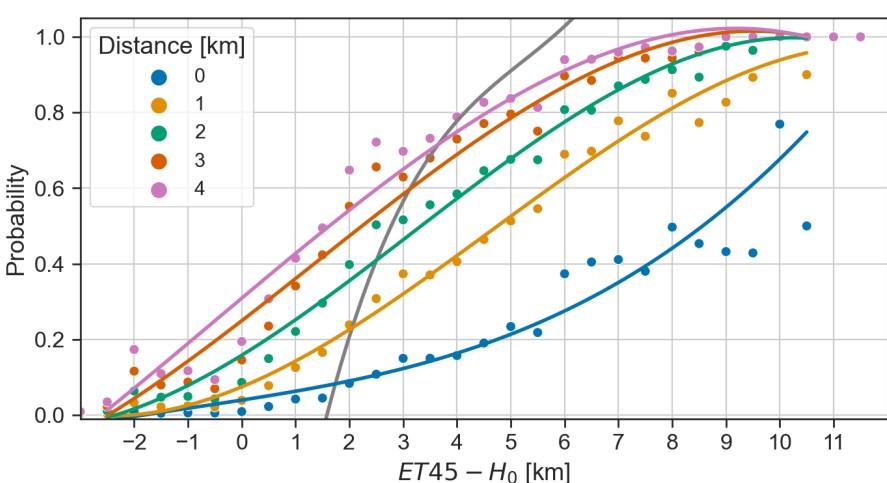

**Figure 16.** Recalibration of POH for the ZRH region using a B19 + HRC 8km/8min filtering method and for different matching distances (colored curves). The grey curve is the original POH calibration.



## 4   Summary, conclusions and outlook

We present a verification of two single polarization radar-based hail metrics used in Switzerland (POH and MESHS) and
suggest a recalibration of POH using a large and dense sample of crowdsourced hail reports gathered through the reporting
function of the MeteoSwiss app. Taking advantage of the high horizontal spatial resolution of the Swiss weather radar network
(1 km) and of the density and precise GPS positioning of the crowdsourced observations, we can use shorter distances than in
previous studies to match our observations with the radar detections.

We make our verification on densely populated regions, only during the daylight hours and during the period when the app
penetration rate was the highest to minimize the number of wrong false alarms from the radar.

As crowdsourced observations can contain intended (jokes) or unintended (misuse) wrong reports, we compare two filtering
methods: one based on radar reflectivity (B19) and another new spatio-temporal clustering approach (ST-DBSCAN) based
solely on the data (HRC). While the B19 has the lowest (best) FAR, we find that the HRC methods systematically lead to a
higher H, CSI, and HSS and that combining both methods further improves H. We conclude that spatiotemporal clustering
can advantageously replace reflectivity as a filter and make the filtered observations fully independent from the radar metrics.
This method could be applied to other types of crowdsourced observations that require filtering, with an appropriate choice of
parameters depending on population density. However, the use of this method also supposes that a majority of people make
correct observations, such that the clustering of wrong observations is limited.

We find lower FAR values (0.3 - 0.7) than in most previous studies verifying POH (Holleman, 2001; Kunz and Kugel, 2015;
Nisi et al., 2016; Puskeiler et al., 2016) using insurance claim data, even though considering a verification grid of smaller
dimensions (4km x 4km). This is because hail of at least 2 cm is required to damage cars or buildings, whereas crowdsourced
reports include all hail sizes and have better area coverage.

The highest CSI (0.37) and HSS (0.52) values are obtained with a threshold of 60% for the ZRH region (80% for the
Swiss100 region), using the B19 + HRC 8km/8min. As CSI and HSS values are almost constant for thresholds between 1%
and 80% for the ZRH region, this confirms the appropriateness of the 80% threshold to derive hail days in Switzerland (Nisi
et al., 2016, 2018; Madonna et al., 2018; NCCS, 2021).

To our knowledge, we present the first complete assessment of the skill of MESHS. We found high FAR values ($> 0.6$) for all
thresholds and methods. The comparison of MESHS and observed hail size at the ground exhibits a large spread. The highest
CSI (0.25) and HSS (0.4), obtained for a 2 cm threshold, are lower than for POH. However, while we focused on regions with
high population density, we acknowledge that it is more difficult to conclude that the largest hailstone over a MESHS grid cell
($1\,km^2$) has indeed been observed than just verifying the presence of hail.

We find that the current calibration of POH does not correspond to a probability, because the range of $1 - FAR_{prob}$ values
does not cover the [0,1] interval. We suggest a recalibration of POH based on the filtered crowdsourced observations which
effectively cover the [0,1] interval by using a wider range of $ET45 - H_0$ values. This recalibration is robust with respect to the
filtering method and explicitly incorporates the matching distance (2 km) in its definition.





We focused on the summer months and the recalibration should hence be further tested during the entire convective season (April to September). The curves for the two regions should be further compared, to determine which one is the most appropriate. Operationally, we also recommend the use of a step function based on our polynomial fit to reflect the related uncertainty and to set it to 0% when $ET45 - H_0 < 0\,km$ to focus on areas where the probability is $> 10 - 20\%$.

A more detailed analysis of the situations leading to misses and false alarms could contribute to further improvements in POH and MESHS. However, their skill will remain limited because they don't capture all the relevant processes involved in hail formation. In that sense, the use of polarimetric radar variables such $Z_{DR}$ or $K_{DP}$ (see e.g. Kumjian, 2013a; Besic et al., 2016) could help identify the hydrometeors species and delineate the updraft strength and horizontal extension (Doswell, 2001; Kumjian, 2013b; Allen et al., 2020). However, individual hailstone trajectories within the updraft (Dennis and Kumjian, 460    2017; Kumjian and Lombardo, 2020) and the microphysical local conditions (Pruppacher and Klett, 2010) which are especially relevant for estimating the sizes of hailstones are much more challenging to observe via operational radars that have to cover large domains and serve many different types of applications at the same time. This explains why the skill of MESHS is below that of POH and, more generally, why estimating the presence of hail is easier than estimating its size.

*Code availability.* The python code used in this study is available on the following github page: https://github.com/jekopp-git/radar_metric_ 465    verifications

**Appendix A: Verification with the upscaling approach for the Swiss100 region**

This section presents the results of the verification of POH and MESHS with the upscaling approach for the Swiss100 region. Compared to the ZRH region, we found slightly higher values for the hit rate and false alarm rate, resulting in slightly lower values for the CSI and HSS, for both radar metrics. For POH, the highest values for the CSI (0.32) and the HSS (0.48) are reached at a threshold of 80% instead of 60%.



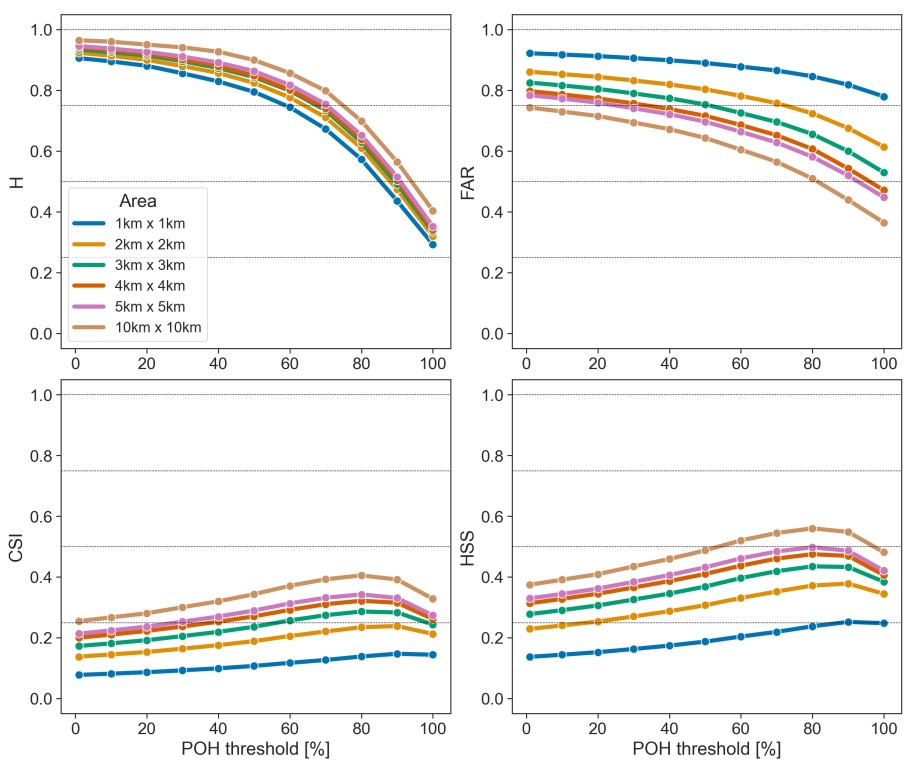

**Figure A1.** POH hit rate (H, left panel) and False alarm ratio (FAR, right panel), Critical Success Index (CSI, bottom left) and Heidke Skill Score (HSS, bottom right) for the Swiss100 region, using the B19 + HRC 8km/8min filter applied on the complete set of observations, stratified by upscaling area (colored curves).

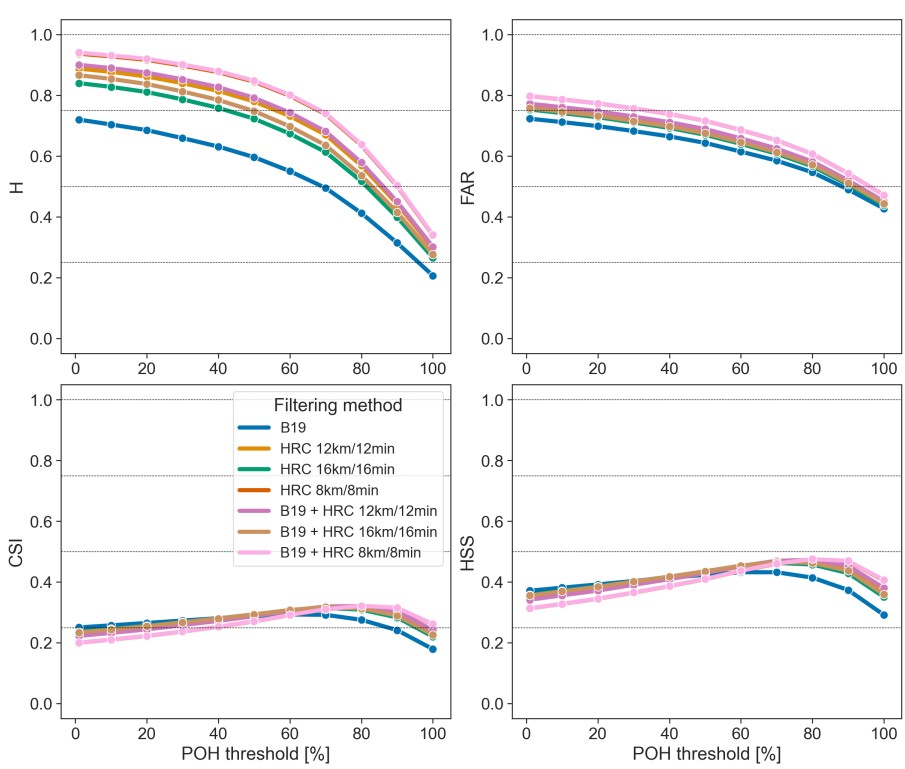

**Figure A2.** POH hit rate (H, left panel) and False alarm ratio (FAR, right panel), Critical Success Index (CSI, bottom left) and Heidke Skill Score (HSS, bottom right) for the Swiss100 region, using the complete set of observations and an upscaling area of 4 km ×4 km, stratified by filtering methods (colored curves).





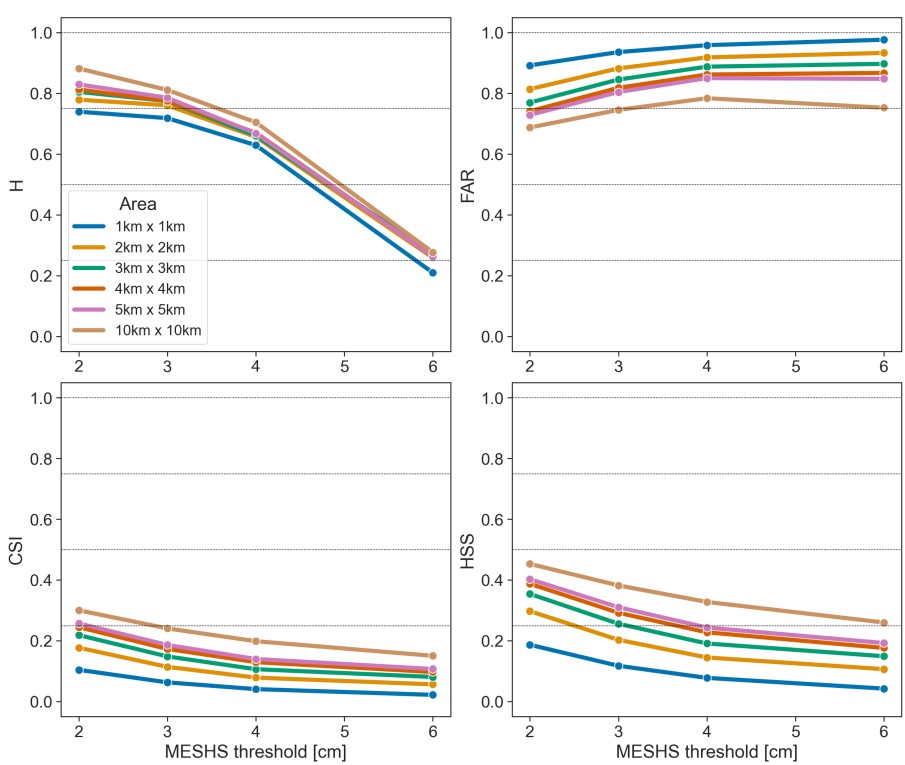

**Figure A3.** MESHS hit rate (H, left panel) and False alarm ratio (FAR, right panel), Critical Success Index (CSI, bottom left) and Heidke Skill Score (HSS, bottom right) for the Swiss100 region, using the B19 + HRC 8km/8min filter applied on the $> 2cm$ set of observations, stratified by upscaling area (colored curves). Each MESHS size threshold is verified against size categories equal to or larger than the threshold.





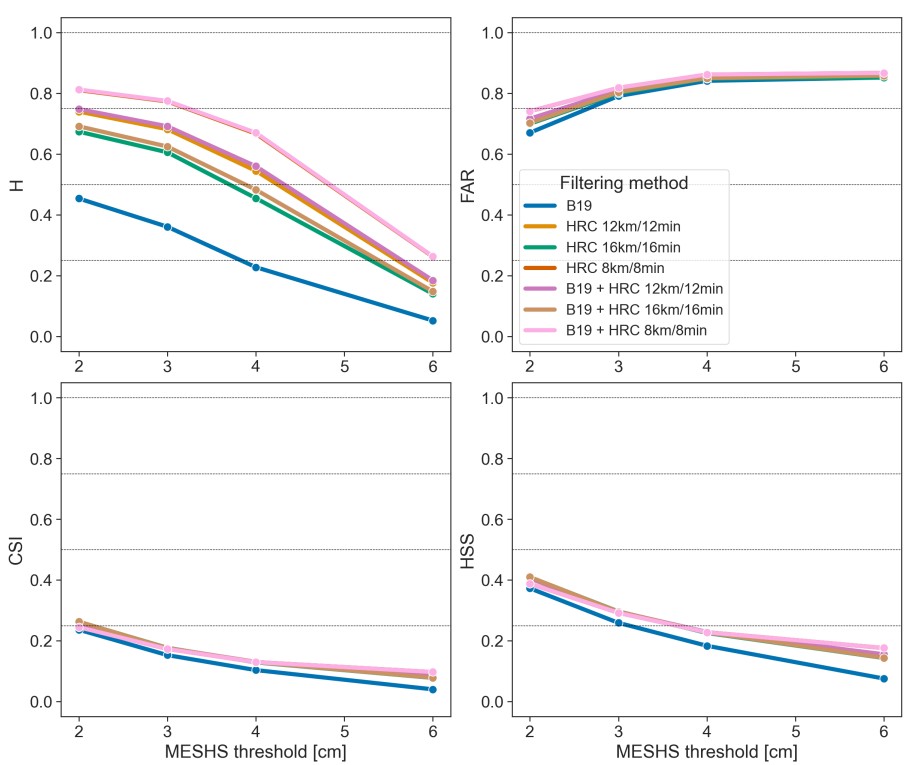

**Figure A4.** MESHS hit rate (H, left panel) and False alarm ratio (FAR, right panel), Critical Success Index (CSI, bottom left) and Heidke Skill Score (HSS, bottom right) for the Swiss100 region, using the $> 2cm$ set observations and an upscaling area of 4 km ×4 km, stratified by filtering methods (colored curves). Each MESHS size threshold is verified against size categories equal to or larger than the threshold.





## Appendix B: Fraction of matches of POH and MESHS

We follow the approach used by Barras et al. (2019) to compute the ratio of crowdsourced reports matching a radar signal within a given spatial and temporal neighborhood. Here, we consider spatial neighborhood from 0 to 4 km (matching distances), and the temporal neighborhood is one day. Each 1-cell of the observations grid is considered a hit if the distance between the report location and the center of a 1-cell of the radar grid is within the matching distance. The fraction of matches is simply the number of hits divided by the total number of observations. Such an approach cannot be used to verify the performance and skill of the radar metric. This is because a single radar cell can be used to match several observation cells.

The fraction of matches is computed as a function of the POH and MESHS thresholds over both the ZRH and Swiss100 regions. We use the complete set of observations (all report sizes) for POH and the $> 2cm$ set of observations for MESHS, keeping only the size categories equal to or larger than the verified MESHS threshold (as in section 3.2). Again, the results for the MESHS 6cm threshold should be taken with care. We assess the sensitivity of the results to the matching distance and to the filtering method, briefly comment on how they differ from the results obtained with the maximum upscaling approach, and compare our results to those of Barras et al. (2019).

Figure B1 shows the results using the B19 + HRC 8km/8min filter for different matching distances, a matching distance of 0 km (blue curve) corresponding to the original grid resolution (and therefore also to an upscaling area of 1 km × 1 km). The fraction of matches improves with increasing matching distance. The largest improvement occurs when passing from the same grid cell (0 km) to 1 km. This is most likely due to some hail observations being close to the border with a neighboring grid cell. The fraction of matches increases only slightly beyond a 2 km buffer and is almost equal to 1 for the lowest POH threshold (1%) on the Swiss100 region. This means that almost all hail observations are within a neighborhood of 2 km of a POH signal. There is a strong decrease in the fraction of matches for POH thresholds of 90% and 100% as with the upscaling approach. For MESHS, the fraction of matches decreases with increasing MESHS thresholds as with the upscaling approach and the decrease for MESHS 6 cm is particularly large.

The fraction of matches of POH and MESHS are lower for the ZRH region (Fig. B1, lower panel) than for the Swiss100 region (Fig. B1, upper panel). This can be explained by the fact that the ZRH region is more densely populated than the Swiss100 region. Some areas of the Swiss100 region might be less populated or even without population, due to the dilation operation on the original cell selection based on a density of 100 people per $km^2$. It is more likely to have wrong reports with a higher population density, and more likely for those reports to be clustered together or with correct ones, thereby lowering the percentage of matches.

As in Barras et al. (2019), we chose a matching distance of 2 km to account for the wind drift of hail and to further compare the results of the different filtering methods on Fig. B2. The results are consistent with those obtained with the upscaling approach (Fig. 9). The largest percentage of matches is reached by the B19 + HRC 8km/8min filter (pink curve), while the smallest percentage of matches is obtained by the B19 filter alone (blue curve). Barras et al. (2019) found a percentage of matches of 54% for the B19 filter and a POH threshold of 1%, using crowdsourced reports over an area that is well covered by the Swiss radar network (between 45.5°N, 5.6°E and 47.9°N, 10.7°E). We find a significantly larger value of 85% for the same



POH threshold and filter over the Swiss100 region (Fig. B2 top left panel, blue curve), and even larger values for HRC filters.

We think that this large difference is due to two elements. First, we restricted our analysis to the summer months, while Barras et al. (2019) used reports from all seasons, including winter and spring, where we found that users reported graupel, sleet and small hail for which the radar reflectivity is most likely below 45 dBZ. Second, Barras et al. (2019) also used a temporal neighborhood of 5 minutes before and after the reporting time to match with POH, whereas we consider daily maximum values of POH (i.e. a temporal neighborhood of 1 day).

For MESHS, Barras et al. (2019) found a percentage of matches of 41% using the B19 filter and for a 2 cm MESHS threshold, but considering only the 1 CHF size category for the verification. Similarly to POH, we find a significantly larger value of 66% for the same MESHS threshold and filter over the Swiss100 region (Fig. 9 top right panel, blue curve), and again larger values for HRC filters. The reasons for this difference are the same as for the POH case (different seasons and temporal neighborhoods).

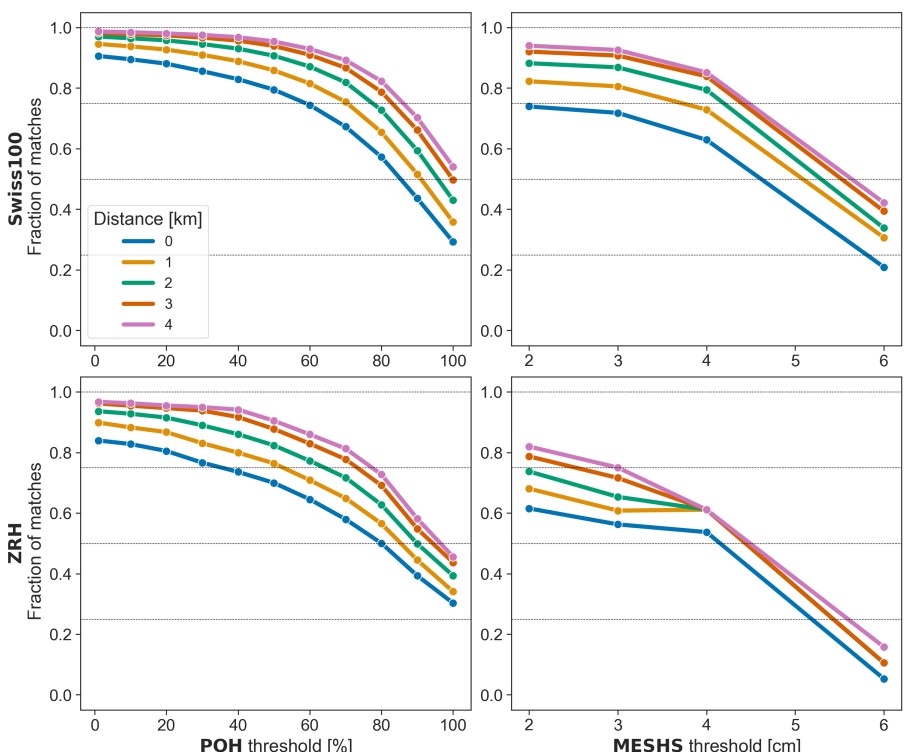

**Figure B1.** Fraction of matches for POH (left column) and MESHS (right column), and for the Swiss100 (top) and ZRH (bottom) regions, using the HRC 8km/8min filter and stratified by matching distance (colored curves). Each MESHS size threshold is verified against size categories equal to or larger than the threshold.



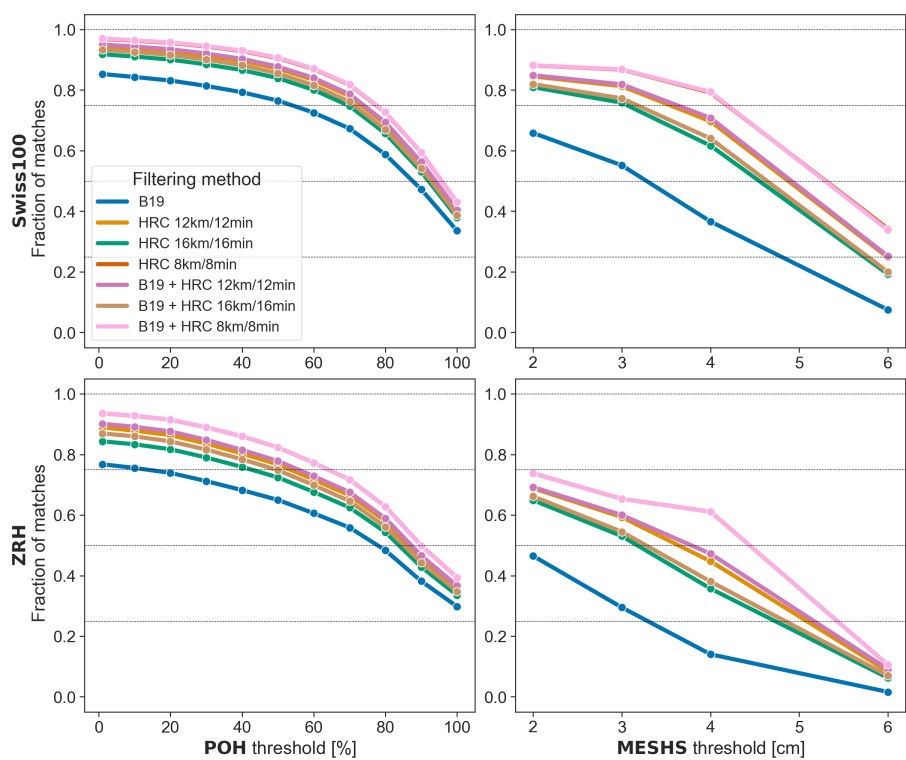

**Figure B2.** Fraction of matches for POH (left column) and MESHS (right column), and for the Swiss100 (top) and ZRH (bottom) regions, using a matching distance of 2 km and stratified by filtering methods (colored curves). Each MESHS size threshold is verified against size categories equal to or larger than the threshold.



## Appendix C: Fraction of matches between POH and automatic hail sensor data

In this section, we look at the fraction of matches between POH and the automatic hail sensor data. We don't apply the upscaling approach to verify FAR, CSI and HSS because of the very small measuring area covered by an individual sensor ($0.2\ m^2$) compared to the spatial resolution of the radar metrics ($1\ km^2$). Due to the localised nature of hail, this would lead to an artificially high number of false alarms. Lainer et al. (2023) noticed that no hail was recorded by an automatic hail sensor located 300 m away from the area where their drone identified more than 18000 hailstones over $600\ m^2$. This implies that even dense networks of hailpads or hail sensors (Federer et al., 1986; Sánchez et al., 2009; Manzato et al., 2022; Kopp et al., 2023) might not capture all hailstreaks. This is not an issue with crowdsourced reports in densely populated areas because the range of vision of a human observer can extend to $100\ m^2$. Besides, more than one observer will likely be present over a grid cell, as we focused on areas with a population density of at least 100 inhabitants per $km^2$. We don't verify MESHS at all with automatic hail sensor data again because hail sensors will likely not be hit by the largest hailstone over a MESHS grid cell due to its small measuring area. For example, Smith and Waldvogel (1989) found that the maximum hailstone diameter observed on hailpads is usually smaller than the largest hailstone found in its vicinity by a human observer due to its small sampling area ($0.1\ m^2$ Towery et al., 1976).

### C1    Data description

Between June 2018 and July 2020 80 automatic hail sensors were installed in the three most hail-prone regions of Switzerland according to the climatology (Nisi et al., 2016, 2018; NCCS, 2021): the Jura (15 sensors) and Napf (38 sensors) north of the Alps and Southern Ticino (27 sensors) south of the Alps (Fig. C1). Each sensor is designed as a Makrolon thermoplastic disc with a diameter of 50 cm, providing a sensing area of approximately $0.2\ m^2$. The sensors convert the vibration caused by the impact of individual hailstones into estimates of kinetic energy and diameter (Löffler-Mang et al., 2011). The sensors provide autonomous and real-time measurements and record the precise time of individual hailstone impacts. For the present study, we consider all impacts with an estimated diameter $\geq$ 5 mm and recorded from 2018-07-09 to 2023-08-31 during the summer months (June, July and August). Following the approach from Kopp et al. (2023), we define a hail event as a series of impacts, each separated by no more than 10 minutes. We further stratify the events by their number of impacts into 3 categories: 2 to 5 (scarce), 6 to 25 (intermediate), and > 25 impacts (dense). We do not consider events with one isolated impact. The maximum reflectivity within a radius of 4km around the sensor at the time of the impact needs to be equal to or higher than 35 dBZ, to avoid recording impacts caused by flying objects or animals (Kopp et al., 2023). This is equivalent to applying the B19 filter, and it results in 608 hail events (358 scarce, 137 intermediate, and 113 dense) comprising 11480 impacts (see Table C1).

### C2    Results

For each hail event, we calculated an average of the largest POH value within a radius of 2 km around the sensor position over the event duration, including the 5 minutes time steps before and after the event to account for storm movement. Table C1 shows the corresponding numbers. We see that the fraction of events having an average POH > 0% is 100% for events with >




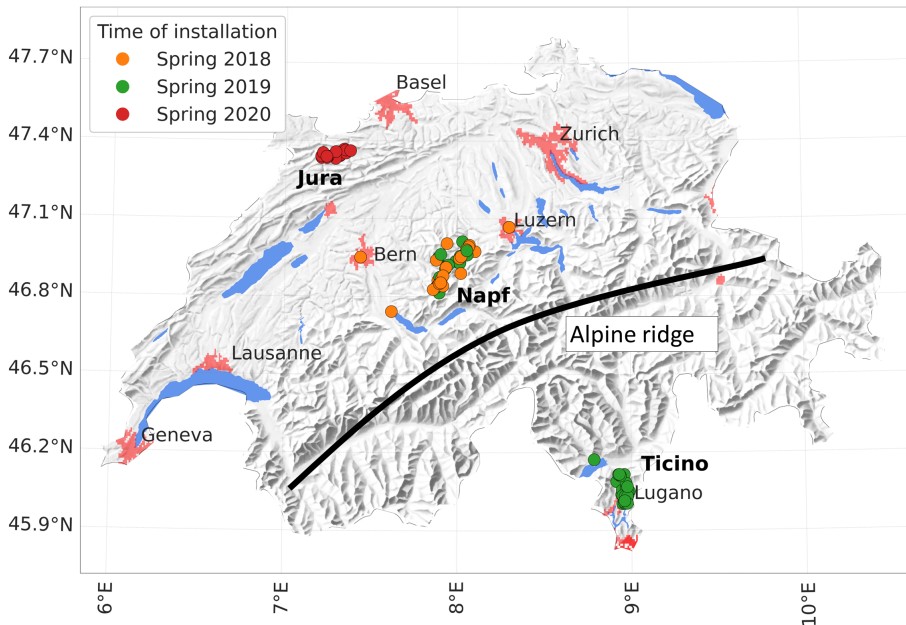

**Figure C1.** Map of Switzerland showing the locations of the 80 sensors according to their installation date in the three hail-prone regions (Jura: 15, Napf: 38, Ticino: 27); red patches show urban areas and the black line denotes the alpine ridge.

**Table C1.** Hail events registered by the hail sensors

|  | 2-5 impacts | 6-25 impacts | >25 impacts | Total |
|---|---|---|---|---|
| number of events | 358 | 137 | 113 | 608 |
| fraction of events with POH > 0% | 73% | 88% | 100% | 81% |

25 impacts, 88% for events having between 6 and 25 impacts, and 73% for events with 2-5 impacts. In total, 81% of events have a matching average POH value. This is lower than the corresponding fraction of matches obtained with the crowdsourced reports (97% for the Swiss100 region with a matching distance of 2 km and the B19 method). However, most events without a

555 POH value belong to the 2-5 impacts event category, and 54% (62) have only 2 impacts. Such low-density events might not be captured by POH due to their low reflectivity or it is possible that the impacts could not be from hail (Kopp et al., 2023).



*Author contributions.* JK: conceptualization; methodology; data preparation and validation; code; statistical and formal analysis; visualisations; writing original draft. AH, UG, OM: conceptualization, review and editing.

*Competing interests.* The authors declare that they have no conflict of interest.

*Acknowledgements.* We thank the Swiss Insurance Company La Mobilière for funding the automatic hail sensors network and making the hail sensor data available. We thank Marcel Belz from MeteoSwiss for his availability and useful help in answering questions related to the crowdsourcing function of the MeteoSwiss app. Jérôme Kopp and Olivia Martius acknowledge support from La Mobilière. Olivia Martius acknowledges support from the Swiss Science Foundation grant number $CRSII5_201792$.



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
