# Peer review of "Verification of weather-radar-based hail metrics with crowdsourced observations from Switzerland"

_EGUsphere, 2024_

## Referee Comment (RC1)

**Review of EGUsphere-2024-729**

This is a review of "A comprehensive verification of the weather radar-based hail metrics POH and MESHS and a recalibration of POH using dense crowdsourced observations from Switzerland" by Kopp et al.

The study presents some useful new findings on hail detection from radar, taking advantage of Switzerland's world-leading observational network for both radar and crowdsourced reports of hail. The important findings include verifications of the often used metrics probability of hail (POH) and maximum expected severe hailstone size (MESHS). A report clustering method that uses only report data and can be used instead of radar-based filtering shows promising results for quality control of dense hail report data. MESHS shows high false alarm rates. An analysis of POH shows that the original version of POH does not correspond with a probability in Switzerland, and the authors show a new calibration of POH.

This is an excellent paper. The results are sound and will be of use to the community who use MESHS, POH and related radar products to infer hail occurrence and/or hail size at the ground. These radar-based metrics are empirical and without well-defined uncertainties. This work, which helps to quantify uncertainties on these measurements in the European domain, is therefore valuable. The paper is highly relevant to AMT.

I suggest that only minor changes and some further explanation of thresholds is required before the paper will be ready for publication.

**General comments**

1. **Threshold choices**: there are thresholds used in the paper (e.g. "populated areas" are those with a population of at least 100 people per km$^2$, reports from a user submitting more than four reports per day are removed, five reports are required for a cluster to be defined, etc) that are given without explanation or justification. I understand the difficulty in choosing a reasonable threshold but it would be helpful to include a discussion of how these thresholds were chosen and whether or not results in the article are sensitive to these choices. I note that the authors have done a great job of justifying and testing other thresholds (such as for EpsD, EpsT, and POH and MESHS thresholds in the verification, for example), so what is required for these other more minor thresholds is simply some reasoning for why the specified values

were chosen.

**Specific comments**

1. Line 33: "because of the nature of hail" - the authors could explicitly name here what the difficult nature is (hailstorms are relatively small and rare, essentially).

2. Line 48: Regarding advection of falling hailstones, a relevant reference may be Ackermann et al 2024 (DOI: 10.5194/amt-17-407-2024).

3. Line 104-105: It would be helpful to mention whether the COSMO runs used here were done for this study or whether operational MeteoSwiss runs were used.

4. Line 117: The definition of maximum column reflectivity (CZC) is missing the fact that the CZC is defined for a given time period (e.g. one radar set of elevation scans or one scan cycle).

5. Figure 4 caption: the Natural Earth populated areas dataset requires a reference.

6. Line 150: I read this as meaning that for a given day and a given point, if the daily maximum radar reflectivity occurred outside 6-22 UTC then the entire day is considered hail free at that point. If this is the case could some valid hailstorms be removed (if for example there was hail during the day but a larger reflectivity was recorded overnight). The authors should consider stating how often grid cells were removed in this way to show any possible impact on the results.

7. Line 161-162 and 167: "hail $< 5$ mm" - since the WMO classifies hail as being at least 5 mm by definition I suggest changing this to "ice particles $< 5$ mm in diameter".

8. Line 262: What the authors refer to here as the POH is also called the success ratio (see e.g. Roebber 2019, DOI: 10.1175/2008WAF2222159.1.)

9. Line 273: CHF is used without a definition here and should be replaced by "Swiss franc".

10. Line 281: It is not really explained why the authors focus on the Zurich region and then present the results for all Switzerland in the appendix. The authors should explain this choice.

11. Line 415: To be picky, "this probability" here refers to a matching distance of 2 km as defined by Equations 2 and 3. I suggest replacing "This probability" with "The probability of observing hail in the neighbourhood of a given point", or similar.

12. Figures 14, 15 and 16: It was not initially clear to me how the probability on the y axis of these plots was calculated. Reading back I see it is $1\text{-}FAR_{prob}$. The authors could consider making this more clear, for example by including $1\text{-}FAR_{prob}$ in the y axis labels.

**Technical corrections**

1. In general I think "weather radar-based" should be written "weather-radar-based".

2. Lines 108-110: The units do not need to be italicised here. This is true elsewhere in the text also, such as on line 119. There is some inconsistency in the writing of units. Units should at all times be non-italicised and with a space between the numeral and the unit.

3. Some in-line citations still have brackets (e.g. on line 21 and in the caption for Figure 2).

4. Line 180: "HRC" is introduced here without definition (the definition comes a few lines later). Acronyms should be defined when first used.

5. Table 2: There is a mix of capitalisations in the column of seasons. Also, "year" is not a season so perhaps the title should be "time period" or similar.

6. Line 384: "restraint" should be "restrained".

7. Figure 14: 'red' and 'green' curves appear as yellow and blue on my display (also affects the text around Line 394).

---

## Referee Comment (RC2)

**Review of EGUsphere-2024-729**

In this paper, the authors exploit a rich dataset of crowdsourced hail reports to examine the validity of radar-based hail metrics employed by Switzerland's National Weather Service (MeteoSwiss), namely the probability of hail at the ground (POH) and the maximum expected severe hailstone size (MESHS), which are crucial for assessing hail presence and estimating its size. Through a new spatiotemporal clustering method (ST-DBSCAN) coupled with radar reflectivity to filter the reports, they conduct a meticulous analysis using various metrics such as Hit Rate, False Alarms Ratio, Critical Success Index, and Heidke Skill Score. They identify shortcomings in the current calibration of POH and suggest a recalibration based on the filtered reports.

The paper is logically structured and very well readable, although the scientificity of the language can be improved here and there. It does not describe groundbreaking science, nor does it scratch on the current state-of-the -art regarding hail detection (e.g., by dual-pol). Nevertheless, in the context of operational meteorology, it contains important information to improve the quality of the hail detection and warning at many meteorological services worldwide. Being in the field myself, I recognize the fact that this kind of publications is unfortunately largely lacking in peer-reviewed scientific literature. Therefore, I strongly recommend publication of the current paper, although some important revisions outlined below are required.

**General comments**

The main issues I see with the paper are listed below. Please provide a solution to all objections and indicate where the text has been adapted accordingly.

- The title is too long. Modify it to two lines maximum. Choose the main message you want to convey and formulate your title accordingly.

- Cited literature is a bit scarce and focused on Swiss publications, some of them being technical reports. I understand the topic is strongly operational, but I suggest widening the literature study a bit in the introduction and conclusion. E.g.: is single-pol hail detection still relevant in times of dual-pol radars being the standard?

- Heidke Skill Score (HSS) is introduced as one of the metrics, but it is not really integrated in the discussion. Either remove HSS from the paper completely, or integrate it better in the discussion.

- "Jumping cells" induced by the finite 5-min radar sampling and the potential high velocity of the hail-producing cells are not mentioned in the paper. Nevertheless, they can jeopardize hail statistics considerably. Or is there an advection correction applied on the radar images? Add some sentences on how this effect can influence your validation results. I suggest adding a figure of the POH 24h overview for the case you describe (June 20, 2021), to have a visual assessment of this effect (if any).

- Appendix C does not fit into the "story" the paper and reads like a separate study. For the reader there is no incentive to read it while reading the main paper. An appendix should always

support the main text, and is not intended to contain a small side-study. I recommend eliminating Appendix C and publish these results elsewhere.

- The authors make several logical choices to limit the wrong false alarms. The scientific motivation for these choices and adopted thresholds is in some cases quite weak or even absent. For example: one retains only the reports between 06:00 and 21:15 UTC, but no justification is given. For this particular choice, one could construct a histogram of all reports versus reporting time and motivate this choice better on the basis of this histogram.

- Similar comment for the definition of the ZRH region. The Swiss100 region is confined with clear criteria based on population density, while the ZRH region is quite arbitrary chosen, in fact without any link with population density. As such, some unpopulated areas, like the Zürichsee, are included in the ZRH region. The authors should either motivate this choice better, or, as an alternative, redo the statistics for the ZRH region with taking a population density criterion to confine the region (similar methodology as the Swiss100 region).

- For POH, a cubic fit is traditionally taken. Given the quite large uncertainty on the exact POH relation (see e.g. Fig. 16), I wonder if such a cubic fit is not overfitting the data points. A goodness-of-fit criterion could be added, and the results for a second order polynomial could be given accordingly. Is the cubic fit justified?

**Specific comments**

L34. Explain "hail hotspots".

L65. Add the total number of inhabitants of Switzerland for reference.

L70. "suspicious reporting patterns": are you referring to the preprocessing explained in the beginning of Section 2.5? If so, add "detailed further in the paper".

L100. Not clear whether the POH is calculated for the individual radars first, and then composed together, *or* that the POH calculation is done on a three-dimensional composite. Please explain briefly on how the POH field from the different radars is calculated.

L104. Spatial and temporal resolution of the COSMO model used? If the temporal resolution of the model is 1h, was there a temporal interpolation to match the radar timestamps? Please explain in more detail.

L125. An additional uncertainty is introduced since a user is only able to report discrete sizes: what should a user report if the hail stone diameter he/she measures is 1.5 mm?

L132. "no one is around to report": gives the impression that the authors assume that everyone in Switzerland has the MeteoSwiss app and will report hail for sure. Hence reformulate.

L134. Does MeteoSwiss send out location-based notifications when hail is approaching to a certain user? Or is this feature not available?

L172. In EU, it is not allowed to retain the complete history of a user ID due to GDPR regulations. Can this be done in Switzerland then?

L174. Why can't a user send more than 4 reports per day? This can be just a very engaged user.

L186. Why 5? Did you do any sensitivity study by varying this number?

L204. 35dBZ could be lacking in the radar data by beam blocking or by "jumping" cells (see above). I miss some nuance here: there can be gaps in the radar data too.

L266. I miss a description in the main text of row 4 of Fig. 7. The "fraction of matches" is not explained in the main text. Please add a short description in the main text, and refer to the bottom row of Fig. 7.

L295. In the introduction, a maximum wind drift of 3 km is cited from literature (L48). Nevertheless 4 km is chosen here. Why? Motivate.

L334. Did you consider eliminating the highest category completely? Hail of this size is extremely rare and the reporting of such sizes even rarer. To me, the majority of these reports seems fake.

L378. Here 2 km is chosen to account for the wind drift: again another choice! Motivate.

L389. The interval (-3km – 12km) is again a quite arbitrary choice. On the contrary, you can look for both the lowest and highest ET45-H0 for which you received a hail report. Please extract these two numbers from your available data and compare it to the current choice.

L405. When there is no echotop 45, I would say ET45 is undefined and not equal to 0! Hence, the argument on L405 seems wrong to me.

L415. So POH maps based on this recalibration should be made not on radar-pixel resolution (1km), but should be provided on 2x2 km$^2$ maps, right?

L442. "first complete assessment of the skill of MESHS" → does this contradict L39-L40 of the introduction?

**Technical comments**

- Typography should be checked and improved throughout the paper, e.g., avoid writing physical units in math font.

L26. Rephrase "and they" … "and they" (split up in several sentences).

L86. Remove "and illustrate the spatio-temporal clustering method in section 2.5.1". Keep the outline at the end of Section 1 limited to the subsections and don't mention subsubsections.

L158-L160. Split in two sentences.

L182. "Noise, it can discover clusters of …" "Noise. It can discover clusters of …" (make two sentences).

L201. (Fig. 6)b) → (Fig. 6b)

L380. pFAR$_{prob}$ → FAR$_{prob}$

L394. "Red" and "green" curves? I see orange and blue…

Caption Fig. 14. "red" and "green"?

---

## Author Comment (AC1)

**Review of EGUsphere-2024-729**

We thank the reviewer for their valuable input, suggestions, and recommendations. We describe below in blue font how we implemented all the points that were raised.

This is a review of "A comprehensive verification of the weather radar-based hail metrics POH and MESHS and a recalibration of POH using dense crowdsourced observations from Switzerland" by Kopp et al. The study presents some useful new findings on hail detection from radar, taking advantage of Switzerland's world-leading observational network for both radar and crowdsourced reports of hail. The important findings include verifications of the often used metrics probability of hail (POH) and maximum expected severe hailstone size (MESHS). A report clustering method that uses only report data and can be used instead of radar-based filtering shows promising results for quality control of dense hail report data. MESHS shows high false alarm rates. An analysis of POH shows that the original version of POH does not correspond with a probability in Switzerland, and the authors show a new calibration of POH.

This is an excellent paper. The results are sound and will be of use to the community who use MESHS, POH and related radar products to infer hail occurrence and/or hail size at the ground. These radar-based metrics are empirical and without well-defined uncertainties. This work, which helps to quantify uncertainties on these measurements in the European domain, is therefore valuable. The paper is highly relevant to AMT. I suggest that only minor changes and some further explanation of thresholds is required before the paper will be ready for publication.

**General comments**

1. Threshold choices: there are thresholds used in the paper (e.g. "populated areas" are those with a population of at least 100 people per km2, reports from a user submitting more than four reports per day are removed, five reports are required for a cluster to be defined, etc) that are given without explanation or justification. I understand the difficulty in choosing a reasonable threshold but it would be helpful to include a discussion of how these thresholds were chosen and whether or not results in the article are sensitive to these choices. I note that the authors have done a great job of justifying and testing other thresholds (such as for EpsD, EpsT, and POH and MESHS thresholds in the verification, for example), so what is required for these other more minor thresholds is simply some reasoning for why the specified values were chosen.

We now explain in more detail the choice of our thresholds in the manuscript. The justifications for the four reports per day and the five reports to form a cluster are given as answers to specific questions from RC2.

Regarding the population threshold of 100 inhabitants per square km, it assumes that if a person can see up to 100 m around, then a hundred people can cover a square km, provided that they are equally distributed in space. Considering a higher threshold would strongly reduce the area and make it too discontinuous.

Section 2.3 has been modified to better explain the choice of the Swiss100 and ZRH regions.

**Specific comments**

1. Line 33: "because of the nature of hail" - the authors could explicitly name here what the difficult nature is (hailstorms are relatively small and rare, essentially).

We made the sentence more explicit as suggested: "Ground-based observations are challenging to gather **because hailstorms are scarce and most of them have a small spatial extension (see, e.g.: Brimelow, 2018)**

2. Line 48: Regarding advection of falling hailstones, a relevant reference may be Ackermann et al 2024 (DOI: 10.5194/amt-17-407-2024).

We thank the referee for making us aware of this recent reference. Following the comments of the second referee, we present a more detailed review of the wind drift effect in the introduction, which includes the findings of Ackermann et al. (2024):

**L48: Looking at 12 severe hailstorms that have occurred in Switzerland, Hohl et al. (2002) found that the best correlation between radar-derived hail kinetic energy and hail damage claims was achieved for wind drifts of 3 to 4.3 km. Analyzing data from seven severe storms in Australia's metropolitan Sydney and Brisbane areas, Schuster et al. (2006) found a wind drift ranging from 2 to 2.8 km. More recently, Ackermann et al. (2024) applied a virtual advection algorithm to 30 hail events that happened between 2010 and 2022 in Australia and found that most events had an estimated wind drift of less than 2 km and that none of them had a wind drift above 4 km. Such values are lower than most distance buffers used in the literature to evaluate the performance of POH.**

3. Line 104-105: It would be helpful to mention whether the COSMO runs used here were done for this study or whether operational MeteoSwiss runs were used.

H0 was retrieved from operational COSMO runs done by MeteoSwiss. We added this information on L104-105.

4. Line 117: The definition of maximum column reflectivity (CZC) is missing the fact that the CZC is defined for a given time period (e.g. one radar set of elevation scans or one scan cycle).

We specified that CZC is defined for a given 5-minute time step (i.e. a full 20-elevation volume scan) on L117.

5. Figure 4 caption: the Natural Earth populated areas dataset requires a reference.

According to the Terms of Use of Natural Earth (https://www.naturalearthdata.com/about/terms-of-use/), only the mention "Made with Natural Earth" is necessary. However, we still added a proper reference to their website.

6. Line 150: I read this as meaning that for a given day and a given point, if the daily maximum radar reflectivity occurred outside 6-22 UTC then the entire day is considered hail free at that point. If this is the case could some valid hailstorms be removed (if for example there was hail during the day but a larger reflectivity was recorded overnight). The authors should consider stating how often grid cells were removed in this way to show any possible impact on the results.

We thank the referee for pointing this out. In fact, this refers to a previous approach that we changed specifically because it could remove valid hailstorms as mentioned by the referee.

What we do in the present study is simply compute the daily maximum of the radar metric between 6:00:00 UTC and 21:00:00 UTC.

We removed the following part "we consider only grid cells where the peak of the hail (defined as the daily maximum of the radar metric) occurred between 6:00:00 UTC and 21:00:00 UTC to ensure enough users are awake to make reports. If the daily maximum is not reached in this interval, the corresponding value for that grid cell is set to 0, as if no hail was predicted."

7. Line 161-162 and 167: "hail < 5 mm" - since the WMO classifies hail as being at least 5 mm by definition I suggest changing this to "ice particles < 5 mm in diameter".

We changed as suggested by the referee.

8. Line 262: What the authors refer to here as the POH is also called the success ratio (see e.g. Roebber 2019, DOI: 10.1175/2008WAF2222159.1.)

We added that **1 - FAR is also called the Success Ratio (Roebber, 2009).**

9. Line 273: CHF is used without a definition here and should be replaced by "Swiss franc".

We replaced CHF with Swiss franc throughout the manuscript.

10. Line 281: It is not really explained why the authors focus on the Zurich region and then present the results for all Switzerland in the appendix. The authors should explain this choice.

We agree that it should be better explained and included this paragraph at the beginning of section 3.1:

**The results for the Swiss100 and ZRH region are comparable and most of the conclusions are identical for both. Therefore, we discuss in details the results for the ZRH region in this section, while the figures for the Swiss100 region are shown in appendix A. We chose to discuss the ZRH region because we think that it has a higher and smoother population density compared to the Swiss100 region, which likely leads to a better estimate of the False alarm ratio.**

11. Line 415: To be picky, "this probability" here refers to a matching distance of 2 km as defined by Equations 2 and 3. I suggest replacing "This probability" with "The probability of observing hail in the neighbourhood of a given point", or similar.

We changed as suggested by the referee.

12. Figures 14, 15 and 16: It was not initially clear to me how the probability on the y axis of these plots was calculated. Reading back I see it is 1-FARprob. The authors could consider making this more clear, for example by including 1-FARprob in the y axis labels.

We changed as suggested by the referee.

**Technical corrections**

1. In general I think "weather radar-based" should be written "weather-radar-based".

We changed accordingly.

2. Lines 108-110: The units do not need to be italicised here. This is true elsewhere in the text also, such as on line 119. There is some inconsistency in the writing of units. Units should at all times be non-italicised and with a space between the numeral and the unit.

The typography was checked and all physical units were written in non-italicised format with a space between the numeral and the unit.

3. Some in-line citations still have brackets (e.g. on line 21 and in the caption for Figure 2).

We corrected the citations.

4. Line 180: "HRC" is introduced here without definition (the definition comes a few lines later). Acronyms should be defined when first used.

We defined the acronym when first used at Line 180.

5. Table 2: There is a mix of capitalisations in the column of seasons. Also, "year" is not a season so perhaps the title should be "time period" or similar.

We changed the title of the column to "Time period" and capitalized the period names.

6. Line 384: "restraint" should be "restrained".

We corrected it accordingly.

7. Figure 14: 'red' and 'green' curves appear as yellow and blue on my display (also affects the text around Line 394).

The curve colors are blue (Swiss100) and orange (ZRH). The text was corrected accordingly.

---

## Author Comment (AC2)

**Review of EGUsphere-2024-729**

We thank the reviewer for their valuable input, suggestions, and recommendations. We describe below in blue font how we implemented all the points that were raised.

In this paper, the authors exploit a rich dataset of crowdsourced hail reports to examine the validity of radar-based hail metrics employed by Switzerland's National Weather Service (MeteoSwiss), namely the probability of hail at the ground (POH) and the maximum expected severe hailstone size (MESHS), which are crucial for assessing hail presence and estimating its size. Through a new spatiotemporal clustering method (ST-DBSCAN) coupled with radar reflectivity to filter the reports, they conduct a meticulous analysis using various metrics such as Hit Rate, False Alarms Ratio, Critical Success Index, and Heidke Skill Score. They identify shortcomings in the current calibration of POH and suggest a recalibration based on the filtered reports.

The paper is logically structured and very well readable, although the scientificity of the language can be improved here and there. It does not describe groundbreaking science, nor does it scratch on the current state-of-the -art regarding hail detection (e.g., by dual-pol). Nevertheless, in the context of operational meteorology, it contains important information to improve the quality of the hail detection and warning at many meteorological services worldwide. Being in the field myself, I recognize the fact that this kind of publications is unfortunately largely lacking in peer-reviewed scientific literature. Therefore, I strongly recommend publication of the current paper, although some important revisions outlined below are required.

**General comments**

The main issues I see with the paper are listed below. Please provide a solution to all objections and indicate where the text has been adapted accordingly.

- The title is too long. Modify it to two lines maximum. Choose the main message you want to convey and formulate your title accordingly.

We wanted our title to be explicit so that a reader could immediately see the salient points of the paper. However, we agree with the referee that a shorter title might also be more understandable. We changed the title as follows: "Verification of weather-radar-based hail metrics with crowdsourced observations from Switzerland".

- Cited literature is a bit scarce and focused on Swiss publications, some of them being technical reports. I understand the topic is strongly operational, but I suggest widening the literature study a bit in the introduction and conclusion. E.g.: is single-pol hail detection still relevant in times of dual-pol radars being the standard?

We added the following paragraphs in the introduction to better explain the relevance of single polarization metrics (L18):

**Weather-radar-based hail metrics can be classified into two categories: single-polarization and dual-polarization (Ryzhkov and Zrnic, 2019). Single polarization metrics are solely based on horizontal reflectivity (ZH) while dual-polarization radar permits the computation of polarimetric variables such as ZDR or KDP (Kumjian, 2013a). Polarimetric variables provide additional observations about the size, shape, and orientation of the hydrometeors**

(Kumjian, 2013a, b; Ortega et al., 2016; Ryzhkov and Zrnic, 2019). Such polarimetric variables can be used in fuzzy-logic classification algorithms to improve the quality of hail detection and size estimation compared to single-polarization metrics (Ryzhkov et al., 2013; Al-Sakka et al., 2013; Ortega et al., 2016; Besic et al., 2016; Steinert et al., 2021).

However, single-polarization metrics are still used operationally by weather services: for example, the Hail Detection algorithms HDA in the United States (Witt et al., 1998) or the maximum expected size of hail (MESH) in Australia (Ackermann et al., 2024). Moreover, as long-time series of single polarization data exist, they permit the computation of long-term hail statistics (see, e.g., Saltikoff et al., 2010; Cintineo et al., 2012; Skripniková and Rezácová, 2014;; Punge and Kunz, 2016; Lukach et al., 2017).

- Heidke Skill Score (HSS) is introduced as one of the metrics, but it is not really integrated in the discussion. Either remove HSS from the paper completely, or integrate it better in the discussion.

We agree with the referee that we should have explained in more detail why we used the HSS in complement to the other scores and that its discussion could be expanded.

Consequently, we made the following changes:

Section 2.5:
- L250: HSS equations were updated (see formulas below)
- L255: **HSS is a forecast skill score based on the proportion correct (PC), which quantifies the accuracy of the radar metric detection compared to a random detection (PC_rand). PC considers the correct rejections (or the nonevents) and thus estimates the ability of the radar metric to correctly predict such nonevents, which happens frequently as hail is rare**.

$$\mathrm{PC} = \frac{a+d}{a+b+c+d} \qquad \mathrm{HSS} = \frac{\mathrm{PC} - \mathrm{PC}_{rand}}{1 - \mathrm{PC}_{rand}} = \frac{2(ad-bc)}{(a+c)(c+d)+(a+b)(b+d)}$$

In sections 4.1 and 4.2, we restructured the presentation of the results such that CSI and HSS are now discussed in a dedicated paragraph, adding some comparison with the literature.

Section 4.1:

**The CSI and HSS values are very close for the six methods incorporating HRC (Fig. 9), while the values for the B19 method alone are visibly lower. The highest CSI (0.37) and HSS (0.52) are reached with the B19 + HRC 8km/8min filter at a threshold of 60\%. Both Holleman (2001) and Nisi et al. (2016) found slightly higher optimal CSI: 0.42 and 0.45, respectively, while Kunz et al. (2015) found a much lower CSI (< 0.2) and HSS (< 0.3). Puskeiler (2016) on the other hand found higher HSS values reaching up to 0.7.**

Section 4.2:

**The CSI and HSS values for MESHS are again extremely similar for the six methods incorporating HRC (Fig. 11), while the values for the B19 method alone are lower. The highest CSI (0.26) and HSS (0.41) are reached with the B19 + HRC 8km/8min filter for a 2 cm threshold. A CSI below 1/3 means that for every hit, the detection metric produces at least 1 false alarm and 1 miss, which can be considered poor detection. However, we note that**

**despite high FAR and low CSI values, HSS systematically remains greater than 0, meaning that MESHS has more skill than random detection.**

- "Jumping cells" induced by the finite 5-min radar sampling and the potential high velocity of the hail-producing cells are not mentioned in the paper. Nevertheless, they can jeopardize hail statistics considerably. Or is there an advection correction applied on the radar images? Add some sentences on how this effect can influence your validation results. I suggest adding a figure of the POH 24h overview for the case you describe (June 20, 2021), to have a visual assessment of this effect (if any).

Currently, no advection correction is applied, but the problem of jumping cells is significantly alleviated by the fast scan strategy of the radar network (see Figure 14 below from Germann et al. 2022, https://doi.org/10.3390/rs14030503):

The antenna rotation speed is comparatively high (3 to 6 rotations per minute), and the scan program uses an interleaved approach with two half-volume scans (blue and yellow), completed every 2.5 min with 20 full sweeps achieved in only 5 minutes. Cartesian radar products such as POH combine the two most recent half-volume scans, and hence take advantage of the interleaved approach. The map of the daily maximum POH for 20 June 2021 (see below) shows no visible jumping cells over Switzerland. Visual inspections of several daily POH and MESHS maps didn't reveal significant jumping cells over Switzerland.

We added the following paragraph discussing jumping cells at Line 119:

**The finite temporal resolution of the radar products can produce striped patterns (or "jumping cells") in case of fast-moving hailstorms (see, for example, Fig. 3 of Kunz et al. 2015). An advection correction routine to interpolate between time steps and obtain a smoothed signal is usually applied (see, e.g., Lukach et al., 2017). Currently, no advection correction is applied to the products from the Swiss radar network, but the problem of jumping cells is significantly alleviated by the fast scan strategy of the radar network (Germann et al., 2022). The antenna rotation speed is comparatively high (3 to 6 rotations per minute), and the scan program uses an interleaved approach with two half-volume scans, completed every 2.5 min with 20 full sweeps achieved in only 5 minutes (see Fig. 14 of Germann et al., 2022). Cartesian radar products such as POH and MESHS combine the two most recent half-volume scans and hence take advantage of the interleaved approach. Visual inspections of several daily POH and MESHS maps did not reveal any visible jumping cells over the regions considered in this study.**

[Figure]

Figure 14 from Germann et al., 2022.

[Figure]

Daily maximum POH for 20 June 2021, from blue (10%) to red (>90%). Country borders are in purple.

- Appendix C does not fit into the "story" the paper and reads like a separate study. For the reader there is no incentive to read it while reading the main paper. An appendix should always support the main text, and is not intended to contain a small side-study. I recommend eliminating Appendix C and publish these results elsewhere.

We wanted to include this complementary analysis with the hail sensors to be exhaustive because we thought that it might be of interest to the hail community.

However, we agree with the referee that it is not part of the main topic of the paper. We removed it from the paper as those results were also published in the PhD thesis of the first author.

- The authors make several logical choices to limit the wrong false alarms. The scientific motivation for these choices and adopted thresholds is in some cases quite weak or even absent. For example: one retains only the reports between 06:00 and 21:15 UTC, but no justification is given. For this particular choice, one could construct a histogram of all reports versus reporting time and motivate this choice better on the basis of this histogram.

We now explain in more detail the choice of our thresholds as per the answers provided below to each specific point mentioned by the present referee (and by the other referee RC1).

Regarding the choice of the daylight hours, the figure below shows the hourly distribution (UTC) of the reports (B19 filter, no regional filter) as an illustration. Less than 6% of the reports are sent between 21:15 UTC and 06:00 UTC. The manuscript already contains 16 figures, so we prefer not to include this histogram, but refer the reader to Fig. 5 of Barras et al. (2019) which shows a distribution of the same shape.

[Figure]

- Similar comment for the definition of the ZRH region. The Swiss100 region is confined with clear criteria based on population density, while the ZRH region is quite arbitrary chosen, in fact without any link with population density. As such, some unpopulated areas, like the Zurichsee, are included in the ZRH region. The authors should either motivate this choice better, or, as an alternative, redo the statistics for the ZRH region with taking a population density criterion to confine the region (similar methodology as the Swiss100 region).

The Swiss100 region has the advantage of being large and of including part of the area where hail is most frequent in Switzerland (see new Figure 4a). However, it also includes some less-populated areas and narrow valleys where hailstorms have less chance to be reported. For this reason, we considered the ZRH region which only has a few square km with less than 100 inhabitants (Figure 4b) such that all hailstorms are likely to be reported. We added a map of the ZRH region to Figure 4, which also illustrates that the area covered by the lakes is relatively small.

- For POH, a cubic fit is traditionally taken. Given the quite large uncertainty on the exact POH

relation (see e.g. Fig. 16), I wonder if such a cubic fit is not overfitting the data points. A goodness-of-fit criterion could be added, and the results for a second order polynomial could be given accordingly. Is the cubic fit justified?

The red (Swiss100) and green (ZRH) curves in the figure below show the second order polynomial fits, along with the existing curves of Figure 14. The correspond coefficients along with their uncertainties are shown in the equations below.

The second order polynomial fits tend to overestimate the probability for low values of ET45-H0 and underestimate the probabilities for large values of ET45-H0, to a larger extent than the cubic fits.

First, it is important to say that the curves of Figure 16 depict different definitions of a probability of hail, each corresponding to the probability of observing hail within a certain distance. In that sense, they are not related to the uncertainty of a single POH definition. For some distances, a second order polynomial or even a linear fit might be a more appropriate choice.

For the chosen 2 km distance, we prefer the cubic fit as it better fits the data and also because it has two inflection points that better capture the cumulated probability distribution shape of the data.

We added the uncertainties on the parameters on Equations 2 and 3, and made the following changes:

L454: The orange and blue curves are cubic fits (Eq. 2 and Eq. 3) based on the data of the respective region and are shown on Eq. 2 and Eq. 3 with the uncertainty of each parameter. We also tested quadratic fits, but their uncertainty was slightly higher. We also preferred the cubic fits because their two inflection points better capture the cumulated probability distribution shape of the data.

L484: Finally, we note that all the curves in Fig. 16 and Fig. 17 are cubic fits shown for comparison and illustration purposes but that we did not test fits of other polynomial orders (quadratic or linear), which might be more appropriate in some cases.

[Figure]

$$y_{ZRH} = 0.1581(\pm 0.0131) + 0.0876(\pm 0.0048) \cdot (ET45 - H_0) + 0.0069(\pm 0.0016) \cdot (ET45 - H_0)^2$$
$$- 0.0007(\pm 0.0001) \cdot (ET45 - H_0)^3$$

$$y_{Swiss100} = 0.0603(\pm 0.0137) + 0.0628(\pm 0.0050) \cdot (ET45 - H_0) + 0.0122(\pm 0.0016) \cdot (ET45 - H_0)^2$$
$$- 0.0098(\pm 0.0001) \cdot (ET45 - H_0)^3$$

$$y_{ZRH} = 0.2063(\pm 0.0161) + 0.1026(\pm 0.0064) \cdot (ET45 - H_0) - 0.0024(\pm 0.0007) \cdot (ET45 - H_0)^2$$

$$y_{Swiss100} = 0.1256(\pm 0.0199) + 0.0829(\pm 0.0079) \cdot (ET45 - H_0) - 0.0002(\pm 0.0008) \cdot (ET45 - H_0)^2$$

**Specific comments**

L34. Explain "hail hotspots".

By hail hotspots, we meant "**the regions where hail is most frequent**" in Switzerland. We replaced hail hotspots with this more explicit sentence and made a reference to the new Figure 4a.

L65. Add the total number of inhabitants of Switzerland for reference.

The Swiss population was approximately **8.9 million people** in 2022. We added this reference.

L70. "suspicious reporting patterns": are you referring to the preprocessing explained in the beginning of Section 2.5? If so, add "detailed further in the paper".

Yes, we added "(**as explained further in section 2.5**)" for clarity.

L100. Not clear whether the POH is calculated for the individual radars first, and then composed together, *or* that the POH calculation is done on a three-dimensional composite. Please explain briefly on how the POH field from the different radars is calculated.

First, the 2-dimensional cartesian products Echotop45 and 50 are calculated based on a composite of the 5 radars. Then a single value of POH is computed for each pixel based on the corresponding ET45 and H0. We included more details in the text as follows:

**The 45 and 50 dBZ echo top height products are used (ET45 and ET50 hereafter) to calculate POH and MESHS, respectively. ET45 and ET50 represent the highest altitude at which a radar reflectivity of at least 45 and 50 dBZ, respectively, is measured. Both echo tops are calculated from a three-dimensional composite of the five radars, and the resulting value is then used to compute POH or MESHS. ET45, ET50, POH, and MESHS are two-dimensional, gridded Cartesian products with a horizontal resolution of 1 km x 1 km and a temporal resolution of 5 minutes.**

L104. Spatial and temporal resolution of the COSMO model used? If the temporal resolution of the model is 1h, was there a temporal interpolation to match the radar timestamps? Please explain in more detail.

**The horizontal resolution of H0 is the same as the echo tops, and its temporal resolution is 1 h. The H0 value of an hour (e.g. 15 UTC) is used to compute the radar metrics for all 5-minute time steps within the hour (15:00 UTC, 15:05 UTC, ..., 15:55 UTC) without interpolation.**

We modified the text accordingly.

L125. An additional uncertainty is introduced since a user is only able to report discrete sizes: what should a user report if the hail stone diameter he/she measures is 1.5 mm?

Currently, no specific instructions are given to the app users on how to report.

L132. "no one is around to report": gives the impression that the authors assume that everyone in Switzerland has the MeteoSwiss app and will report hail for sure. Hence reformulate.

We reformulated as follows: "**because no user of the app is around to report**".

L134. Does MeteoSwiss send out location-based notifications when hail is approaching to a certain user? Or is this feature not available?

MeteoSwiss automatically generates thunderstorm warnings for the whole of Switzerland. These go to the authorities via a dedicated platform and to the public via the app. There is no specific warning for hail (yet).

L172. In EU, it is not allowed to retain the complete history of a user ID due to GDPR regulations. Can this be done in Switzerland then?

As a Swiss-based entity, we are subject to the Federal Act on Data Protection of the Swiss Confederation.

L174. Why can't a user send more than 4 reports per day? This can be just a very engaged user.

We assume that a user will send a single report per hailstorm and that it would be very unlikely for a user to encounter more than 4 different hailstorms during the same day. It could be the case for people who chase storms, but this represents very, very few users, compared to potential spamming.

We added the following sentence at L174 to explain this filter:
**This additional filter was added to remove potential spamming cases, assuming that a user will send a single report per hailstorm and that it would be very unlikely for a user to encounter more than 4 different hailstorms during the same day.**

L186. Why 5? Did you do any sensitivity study by varying this number?

We agree that this choice should be better explained. We added the following paragraph at L186:
**During preliminary tests of the clustering algorithm (not shown), we noticed that increasing the minimum number of reports to form a cluster was equivalent to decreasing EpsD and EpsT, as both resulted in fewer reports being clustered. Hence, we decided to fix this minimum number to a reasonable value and only vary EpsD and EpsT. In our case, we require at least 5 grouped reports to form a cluster because it is large enough to be confident that hail occurred (5 different users reporting hail independently) and because**

**values of 10 or more would be too restrictive with respect to the density of users according to our tests.**

L204. 35dBZ could be lacking in the radar data by beam blocking or by "jumping" cells (see above). I miss some nuance here: there can be gaps in the radar data too.

We agree that potential gaps in the radar data should be mentioned. The CZC data used for the B19 filter is also a composite of the five radars. So within Switzerland, each pixel is usually seen by at least 2 or more radars from different directions, hence minimizing the potential gaps in the radar data.

We modified it as follows:

**Potential gaps in the CZC radar product can impact the B19 filter. However, as we work with composite from the five radars, each pixel within Switzerland is usually seen by at least two or more radars from different directions, hence minimizing such gaps. Assuming that the B19 filter has very limited gaps,** and as it is  unlikely that hail occurs below 35 dBZ, those reports are likely intended false reports or non-intended errors.

L266. I miss a description in the main text of row 4 of Fig. 7. The "fraction of matches" is not explained in the main text. Please add a short description in the main text, and refer to the bottom row of Fig. 7.

We moved the description of the fraction of the matches from Appendix B to section 2.6 and added a reference to Fig. 7.

L295. In the introduction, a maximum wind drift of 3 km is cited from literature (L48). Nevertheless 4 km is chosen here. Why? Motivate.

We agree with the referee that the choice of the matching distance should be better explained.

First, we present in more details the existing findings regarding the wind drift in the introduction, including an additional reference suggested by RC1:

**L48: Looking at 12 severe hailstorms that have occurred in Switzerland, Hohl et al. (2002) found that the best correlation between radar-derived hail kinetic energy and hail damage claims was achieved for wind drifts of 3 to 4.3 km. Analyzing data from seven severe storms in Australia's metropolitan Sydney and Brisbane areas, Schuster et al. (2006) found a wind drift ranging from 2 to 2.8 km. More recently, Ackermann et al. (2024) applied a virtual advection algorithm to 30 hail events that happened between 2010 and 2022 in Australia and found that most events had an estimated wind drift of less than 2 km and that none of them had a wind drift above 4 km. Such values are lower than most distance buffers used in the literature to evaluate the performance of POH.**

Then we now explain our choice as follows on L295:

**As mentioned in the introduction, most previous studies implicitly incorporated this wind drift effect because the spatial resolution of their observations (or radar detections) was coarser (10 km or more). We would like to use a distance buffer that correctly reflects the wind drift effects. Considering that most studies reported wind drift distances below 4 km, while Hohl et al. (2002) documented values reaching up to 4.3 km for hailstorms in Switzerland, we selected a 4 km x 4 km area as a compromise to further assess the skill of POH.**

L334. Did you consider eliminating the highest category completely? Hail of this size is extremely rare and the reporting of such sizes even rarer. To me, the majority of these reports seem fake.

The Tennis Ball category is supposed to correspond to hailstones larger than 7 cm. While we agree that such large hailstones are not common, we have observed them on some occasions (e.g., 28 June 2021) and cannot completely rule them out. Also, the different filters efficiently removed the most implausible cases of such large hailstones, and we ended up with very few reports of this category, as shown by the numbers in the last paragraph of this section.

L378. Here 2 km is chosen to account for the wind drift: again another choice! Motivate.

In this case, we wanted to have the shortest possible distance so that the information is still useful on a local scale while having a well-defined cumulative probability distribution of the data. The 2 km matching distance satisfied those criteria. Other choices of the matching distance are possible depending on the user's needs. However, we would not recommend going below 2 km, which corresponds to the average wind drift value reported in Ackermann et al. 2024, the most recent study analyzing wind drift effects.

We added this explanation at the end of section 3.4

L389. The interval (-3km – 12km) is again a quite arbitrary choice. On the contrary, you can look for both the lowest and highest ET45-H0 for which you received a hail report. Please extract these two numbers from your available data and compare it to the current choice.

The choice of the interval is based on the distribution of the ET45-H0 values. The lowest and highest values of ET45-H0 for reports filtered with the B19 + HRC 8km/8min filter over the Swiss100 region are -4.563km and 11.912km respectively. Large negative values of ET45-H0 correspond to cases where ET45 is undefined and set to 0, with a high altitude H0. Hence, values of H0 above 3000m without reflectivity correspond to a clear sky and high-temperature conditions, where hail is not expected.

We added this explanation on L389.

L405. When there is no echotop 45, I would say ET45 is undefined and not equal to 0! Hence, the argument on L405 seems wrong to me.

We agree that when the maximum reflectivity is below 45 dBZ, the echotop 45 is not defined. In this particular case, ET45-H0 is used as a proxy for the zone where hail can grow. For a given H0, as ET45 decreases, ET45-H0 approaches 0 and then can become negative, corresponding to a decreasing storm intensity and probability of hail. We chose to set undefined ET45 equal to 0 for coherence as it results in smaller values of ET45-H0 and a smaller probability of hail.

We wrote explicitly that we set ET45 to 0 when it is not defined.

L415. So POH maps based on this recalibration should be made not on radar-pixel resolution (1km), but should be provided on 2x2 $km_2$ maps, right?

This is a relevant question, thank you. The recalibration of POH is based on the ET45-H0 values of 1km radar pixel so each 1km pixel will have its own recalibrated value of POH. Two adjacent

1km radar pixels have different 2km neighborhoods so they can have different values of the recalibrated POH. We added this explanation to the manuscript:

**The recalibrated POH incorporates the matching distance in its definition but still has the spatial resolution of the radar grid. Indeed, two adjacent grid cells have different 2 km neighborhoods, so they can have different values of the recalibrated POH.**

L442. "first complete assessment of the skill of MESHS" → does this contradict L39-L40 of the introduction?

The verification of MESHS presented by Joe et al. (2004) is qualitative, while the one presented by Barras et al. (2019) only consists of the hit rate of MESHS. Our assessment is based on a contingency table, which includes false alarms.

We changed the sentence on L442 as follows to make it more explicit: "**We present the first assessment of the skill of MESHS using a contingency table of detections and observations**."

We also mentioned the details of the verification by Joe et al. (2004) and Barras et al. (2019) in the introduction: Treloar (1997) used 27 hailstorms in the area of Sydney to propose the initial version of MESHS and Joe (2004) verified it **qualitatively** with a single day of data in Australia.

To our knowledge, only Barras (2019) verified MESHS **by calculating the percentage of matches between MESHS values and crowdsourced observations but did not quantify the potential false alarms of MESHS.**

**Technical comments**

- Typography should be checked and improved throughout the paper, e.g., avoid writing physical units in math font.

The typography was checked and all physical units were written in non-italicised format with a space between the numeral and the unit.

L26. Rephrase "and they" ... "and they" (split up in several sentences).

We split it into two sentences.

L86. Remove "and illustrate the spatio-temporal clustering method in section 2.5.1". Keep the outline at the end of Section 1 limited to the subsections and don't mention subsubsections.

We removed the mention of the subsubsection in the outline.

L158-L160. Split in two sentences.

We split it into two sentences.

L182. "Noise, it can discover clusters of ..." "Noise. It can discover clusters of ..." (make two sentences).

We split it into two sentences

L201. (Fig. 6)b) → (Fig. 6b)

We corrected it accordingly.

L380. pFAR$_{prob}$ → FAR$_{prob}$

We corrected it accordingly.

L394. "Red" and "green" curves? I see orange and blue...
Caption Fig. 14. "red" and "green"?

The curve colors are blue (Swiss100) and orange (ZRH). The text was corrected accordingly.